# Slowly Reducible Genetically Encoded Green Fluorescent Indicator for In Vivo and Ex Vivo Visualization of Hydrogen Peroxide

**DOI:** 10.3390/ijms20133138

**Published:** 2019-06-27

**Authors:** Oksana M. Subach, Tatiana A. Kunitsyna, Olga A. Mineyeva, Alexander A. Lazutkin, Dmitri V. Bezryadnov, Natalia V. Barykina, Kiryl D. Piatkevich, Yulia G. Ermakova, Dmitry S. Bilan, Vsevolod V. Belousov, Konstantin V. Anokhin, Grigori N. Enikolopov, Fedor V. Subach

**Affiliations:** 1National Research Center “Kurchatov Institute”, Moscow 123182, Russia; 2Moscow Institute of Physics and Technology, State University, Moscow 123182, Russia; 3P.K. Anokhin Institute of Normal Physiology, Moscow 125315, Russia; 4MIT Media Lab, Massachusetts Institute of Technology, Cambridge, MA 02139-4307, USA; 5Shemyakin-Ovchinnikov Institute of Bioorganic Chemistry, Moscow 117997, Russia; 6Pirogov Russian National Research Medical University, Moscow 117997, Russia; 7Institute for Advanced Brain Studies, Lomonosov Moscow State University, Moscow 119991, Russia; 8Center for Developmental Genetics and Department of Anesthesiology, Stony Brook University, Stony Brook, NY 11794, USA

**Keywords:** genetically encoded hydrogen peroxide indicator, gamma-irradiation, in vivo imaging, ex vivo, brain, fixation, redox

## Abstract

Hydrogen peroxide (H_2_O_2_) plays an important role in modulating cell signaling and homeostasis in live organisms. The HyPer family of genetically encoded indicators allows the visualization of H_2_O_2_ dynamics in live cells within a limited field of view. The visualization of H_2_O_2_ within a whole organism with a single cell resolution would benefit from a slowly reducible fluorescent indicator that integrates the H_2_O_2_ concentration over desired time scales. This would enable post hoc optical readouts in chemically fixed samples. Herein, we report the development and characterization of NeonOxIrr, a genetically encoded green fluorescent indicator, which rapidly increases fluorescence brightness upon reaction with H_2_O_2_, but has a low reduction rate. NeonOxIrr is composed of circularly permutated mNeonGreen fluorescent protein fused to the truncated OxyR transcription factor isolated from *E*. *coli.* When compared in vitro to a standard in the field, HyPer3 indicator, NeonOxIrr showed 5.9-fold higher brightness, 15-fold faster oxidation rate, 5.9-fold faster chromophore maturation, similar intensiometric contrast (2.8-fold), 2-fold lower photostability, and significantly higher pH stability both in reduced (p*K_a_* of 5.9 vs. ≥7.6) and oxidized states (p*K_a_* of 5.9 vs.≥ 7.9). When expressed in the cytosol of HEK293T cells, NeonOxIrr demonstrated a 2.3-fold dynamic range in response to H_2_O_2_ and a 44 min reduction half-time, which were 1.4-fold lower and 7.6-fold longer than those for HyPer3. We also demonstrated and characterized the NeonOxIrr response to H_2_O_2_ when the sensor was targeted to the matrix and intermembrane space of the mitochondria, nucleus, cell membranes, peroxisomes, Golgi complex, and endoplasmic reticulum of HEK293T cells. NeonOxIrr could reveal endogenous reactive oxygen species (ROS) production in HeLa cells induced with staurosporine but not with thapsigargin or epidermal growth factor. In contrast to HyPer3, NeonOxIrr could visualize optogenetically produced ROS in HEK293T cells. In neuronal cultures, NeonOxIrr preserved its high 3.2-fold dynamic range to H_2_O_2_ and slow 198 min reduction half-time. We also demonstrated in HeLa cells that NeonOxIrr preserves a 1.7-fold ex vivo dynamic range to H_2_O_2_ upon alkylation with N-ethylmaleimide followed by paraformaldehyde fixation. The same alkylation-fixation procedure in the presence of NP-40 detergent allowed ex vivo detection of H_2_O_2_ with 1.5-fold contrast in neuronal cultures and in the cortex of the mouse brain. The slowly reducible H_2_O_2_ indicator NeonOxIrr can be used for both the in vivo and ex vivo visualization of ROS. Expanding the family of fixable indicators may be a promising strategy to visualize biological processes at a single cell resolution within an entire organism.

## 1. Introduction

Small signaling molecules define and coordinate changes in the brain that mediate experience-dependent neuronal plasticity or underlie neurological disorders. Ample experimental evidence indicates that H_2_O_2_ plays a crucial role as a regulatory molecule in the brain in health and disease [1,2,3]. Depending on the intracellular concentration, H_2_O_2_ can act as a signaling molecule or can cause oxidative stress followed by adaptation or apoptosis of the affected cell. In the nervous system, H_2_O_2_ is thought to act as a neuromodulator and be reported to be involved in signaling, synaptic plasticity, long-term potentiation, and formation of long-term memory [4].

Genetically encoded fluorescent protein indicators for H_2_O_2_ allow real-time visualization of changes in H_2_O_2_ concentration within individual mammalian cells [5,6]. Several available recombinant indicators of H_2_O_2_, based on different H_2_O_2_-sensing moieties, have been proposed. Of those, indicators of the HyPer family are similar to Orp1-roGFP in their sensitivity to H_2_O_2_ but have faster reaction rate than the latter [7,8,9]. A new highly-sensitive family of yeast peroxiredoxin-based indicators has recently been reported [10]; however, it is not functional in mammalian cells. 

While HyPer sensors can reliably report increases in H_2_O_2_ concentration, their fluorescence response in mammalian cells is readily reversible and once H_2_O_2_ is eliminated, their fluorescence recovers to the original reduced state within a few minutes. The short recovery time of the HyPer indicators is useful for monitoring in vivo cellular processes that occur with fast kinetics; however, the same feature impedes their utilization for following the processes that have slow kinetics and provides less time for fixation which is necessary for the ex vivo visualization of H_2_O_2_ in chemically fixed samples. Hence, it will be highly useful to engineer a slowly reducible indicator for H_2_O_2_. It will be also beneficial if such an indicator would preserve its fluorescence under chemical fixation conditions without altering the ratio of the reduced and oxidized states upon chemical fixation. Such slowly reducible indicators could enable H_2_O_2_ visualization when direct real time measurements are not possible, for instance whether animals are exposed to various exogenous stimuli or when trained in complex behavioral tasks. Additional requirements for such slowly reducible sensors of H_2_O_2_ are brightness, dynamic response range, and stability to pH changes (which can compromise interpretation of the experimental data). 

Evidence from the molecular evolution of GFP-like proteins indicates that the properties of an original template primarily predetermine the properties of the derived fluorescent proteins (FPs) [11]. Therefore, we considered using a monomeric yellow-green fluorescent protein mNeonGreen, which has the highest brightness among the most commonly used green and yellow FPs [12] and has higher pH stability than enhanced yellow fluorescent protein (EYFP), employed as a fluorescence moiety in most of the HyPer versions. Hence, it would be beneficial to generate a bright and pH-stable slowly reducible green H_2_O_2_ indicator using mNeonGreen protein to preserve the advantageous properties of the template.

Yet another issue critical for monitoring H_2_O_2_ production and action is preservation of the reduction/oxidation status of the sensors and, in particular, preventing nonspecific oxidation of the sensors. During the standard procedure of tissue fixation using paraformaldehyde as well as during tissue sample storage, cysteine residues of proteins are rapidly oxidized. One strategy to prevent this process is selective modification of the thiol groups of cysteine residues in the reduced state. The possibility of cysteine residues’ stabilization in a reduced state in proteins in vivo was previously demonstrated for bacteria by alkylation with alpha-methylstyrene a transcription factor OxyR from *E. coli*, which is used in HyPer indicators as the H_2_O_2_ sensing domain [13]. The cysteine residues of the redox indicators Orp1-roGFP and Grx1-roGFP expressed in *Drosophila* larvae can also be preserved in a non-oxidized state using alkylation with N-ethylmaleimide (NEM) and subsequent fixation [7,8,9].

NEM and iodoacetamide (IAM) are the most common alkylating reagents, with NEM reacting with thiols more rapidly than IAM [14,15]. In addition, the NEM alkylation reaction is less pH-dependent. However, NEM can act nonspecifically at an alkaline pH: in addition to cysteine thiols, it also reacts with the side chains of lysine and histidine [16]. Furthermore, IAM has been shown to have higher cytotoxicity than NEM, such that the treatment of cells with IAM derivatives, in contrast to NEM derivatives, leads to endoplasmic reticulum stress [17] as well as apoptosis [14]. Thus, NEM is better suited as an alkylating agent for biological samples.

In this study, we addressed the requirements for an efficient and pH-stable H_2_O_2_ sensor by generating a circularly permutated version of mNeonGreen fluorescent protein and, by fusing it to a truncated OxyR transcription factor, engineering a bright and pH-stable green fluorescent recombinant indicator for H_2_O_2_ with a very slow reduction rate. The indicator, which we designated as NeonOxIrr, demonstrated key characteristics in vitro that are similar or better than those of HyPer3. NeonOxIrr showed a high dynamic range and slow reduction phenotype when exposed to external H_2_O_2_ in the cytosol of bacterial, HEK293T, HeLa and neuronal cells or in various cellular organelles, ranging from nucleus and cell membranes to the matrix and intermembrane space of mitochondria. We demonstrated that NeonOxIrr could reveal endogenous reactive oxygen species (ROS) production in HeLa cells. In contrast to HyPer3, in HEK293T cells, NeonOxIrr could visualize other optogenetically produced ROS that are not H_2_O_2_ species. We also developed an alkylation-fixation method for the NeonOxIrr indicator that preserved its high ex vivo response to external H_2_O_2_ in bacterial and mammalian HeLa cells. The same alkylation-fixation procedure in the presence of detergent reagent allowed ex vivo registering external H_2_O_2_ in neuronal cultures and in the cortex of the mouse brain. Finally, using the NeonOxIrr indicator, we attempted to detect endogenous ROS production either in the mouse cortex or hippocampus after its exposure to γ-irradiation. Thus, NeonOxIrr is the first example of an H_2_O_2_ indicator that allows ex vivo visualization of the H_2_O_2_ signals. 

## 2. Results 

### 2.1. Developing a Slowly-Reducible Green Indicator for hydrogen peroxide in a Bacterial System

To develop a slowly reducible indicator for hydrogen peroxide, we fused the fluorescent mNeonGreen protein with a truncated H_2_O_2_-sensing module of OxyR and performed several rounds of optimization using directed evolution in a bacterial system [11]. The mNeonGreen protein was chosen as an original template for the fluorescent part of the sensor because of its increased brightness and pH stability compared to available FPs of the green-yellow spectral range [12]. Based on the amino acids’ alignment for mNeonGreen and EYFP, we identified the circular permutation site to be between residues 144 and 145, analogous to that in circularly permutated YFP protein used in HyPers (green arrow, ESI Appendix A; here and below the amino acids’ enumeration follows that for EGFP) [5]. As a sensory moiety, we used a truncated version of the OxyR protein from *E. coli* analogous to that in HyPer [5]. We then generated rational libraries for the H_2_O_2_ indicator, composed of the fluorescent and sensory domains of mNeonGreen and OxyR, respectively, with randomized 3 amino acid long linkers between those domains (Figure 1a,b, ESI Appendix A) [18]. We screened 2 × 10^4^ clones of mutants from these libraries on Petri dishes under fluorescent stereomicroscope, followed by analysis in bacterial suspension and as pure proteins. We found fluorescent variants that responded to treatment with 10–100 mM H_2_O_2_ with a 5%–40% increase in fluorescence brightness. 

The best variants selected from the initial rational mutagenesis library were subjected to 6 sequential rounds of random mutagenesis, each followed by screening of 2 × 10^4^ mutants on Petri dishes and subsequent analysis of 10–20 and 1–2 clones in bacterial suspension, and as pure proteins, respectively. The number of rounds was determined by reaching a plateau from round to round for characteristics such as brightness and contrast. To select versions of indicator with low reduction rates, we sprayed H_2_O_2_ on the dishes with bacterial colonies and 20 and 180 min later searched for clones with the highest contrast within this time interval. The final version of the indicator was designated as NeonOxIrr. This version differed from the original template by 13 mutations, including those in the linkers (ESI Appendix A). 4, 3 and 6 mutations were located in the fluorescent mNeonGreen, OxyR and linker parts, respectively. All mutations in the fluorescent part were external to the β-barrel fold and were not expected to dramatically affect the properties of the GFP chromophore. The P99S mutation in the OxyR domain was assumed as a plausible candidate to reduce the reduction rate upon the oxidation of the NeonOxIrr indicator.

### 2.2. In vitro Characterization of NeonOxIrr 

We further characterized the main spectral and chemical properties of the NeonOxIrr indicator in vitro and compared them to those of HyPer3 and EGFP proteins (Table 1). In the oxidized state, NeonOxIrr_ox_ exhibited fluorescence with excitation/emission peaks at 508/520 nm (Figure 1c). Its brightness in the oxidized state was 5.9- and 1.7-fold higher than that of HyPer3_ox_ and mEGFP, respectively. 

The HyPer3 and mEGFP characteristics are measured in our lab using the same conditions as for NeonOxIrr. Characteristics for proteins in an oxidized state were measured after oxidation with an equimolar amount of hydrogen peroxide (20–200 µM). ND, not determined. NA, not applicable. ^1^ Determined by alkaline denaturation in 1N NaOH. ^2^ Relative to mEGFP in PBS buffer (QY is 0.60); quantum yield for the reduced state was estimated according to absorbance and fluorescence changes in response to a 200 µM H_2_O_2_ addition to bacterial lysates using the following equation: QY^Red^ = QY^Ox^ × (Abs^Ox^/Abs^Red^)/(Fluor^Ox^/Fluor^Red^).^3^ pKas for the reduced state were measured in the presence of 14 mM DTT. ^4^ Impact of a pH change of ±0.5 units to the contrast was calculated using the following equation: 100 × |Fluor ^Red at pH ± 0.5^ – Fluor ^Red at pH^|/|Fluor^Ox at pH^ - Fluor^Red at pH^|, where Fluor ^Red at pH ± 0.5^, Fluor ^Red at pH^, and Fluor^Ox at pH^ are the fluorescence intensities of the indicator in the reduced and oxidized states at pH values of pH ± 0.5 and the original pH. Variations of the maximal values of pH impacts for the indicated pH range are shown. ^5^ Maturation was performed on the bacterial lysate in the presence of 28 mM DTT to prevent cysteine oxidation as well as 30 µg/mL kanamycin and 25 µg/mL chloramphenicol antibiotics to block protein synthesis. ^6^ Time for bleaching from an initial emission rate of 1000 photons/s down to 500 photons/s was calculated using a 63 × oil objective lens and 1 mg/mL pure oxidized proteins (in 1 × PBS) in oil (see details in ESI Appendix A). ^7^ Bacterial suspensions were titrated with H_2_O_2_ (0–200 µM), and maximally achievable fluorescence contrasts are shown. 

To estimate the impacts of the quantum yield and extinction coefficient on the fluorescence contrast of the NeonOxIrr indicator, these parameters were compared in the reduced and oxidized states. Oxidation of NeonOxIrr indicator with H_2_O_2_ did not change the fluorescence quantum yield of its chromophore despite a large observed change in fluorescence (Table 1). Hence, the change in the extinction coefficient of the NeonOxIrr protein should be the major contributor to the dynamic range of NeonOxIrr. Indeed, we observed a reduction in absorbance at 346 nm and an increase in absorbance at 508 nm upon oxidation with H_2_O_2_ (ESI Appendix A). This suggests that the oxidation of NeonOxIrr is accompanied by the transition of one form of the chromophore (with absorbance at 346 nm) into another form (with absorbance at 508 nm). An opposite transition was observed upon the reduction of the indicator. We could not detect fluorescence of the NeonOxxIrr form with absorbance at 346 nm under a fluorescent microscope because of its substantially lower brightness compared with the main fluorescent form with absorbance at 508 nm.

We next investigated the effect of pH on the key characteristics of NeonOxIrr. The physiological pH in different cell organelles varies from 5.0 in lysosomes to 7.5 in the cytosol [19] or to even higher values in the mitochondrial matrix [20] and can undergo significant fluctuations in response to various stimuli [21]. Therefore, we studied the pH dependence of the fluorescence and dynamic range of the NeonOxIrr indicator, comparing them with the corresponding characteristics for HyPer3. NeonOxIrr had the same p*K_a_* value (5.9) in the oxidized and reduced states, which was substantially lower than those for HyPer3 (p*K_a_* ≥ 7.6 and p*K_a_* ≥ 7.9, respectively; Figure 1d and ESI Appendix A). The dynamic range of NeonOxIrr was constant in the pH range of 5.5–7.5 and increased upon acidification to pH 5.0 or alkalization to pH 8.0. In contrast, HyPer3 showed a high pH dependence of its dynamic range in the range of pH 5.5–7.5. 

To systematically estimate the pH sensitivity of the purified NeonOxIrr, we normalized its fluorescence changes caused by alkalization/acidification on +/−0.5 units to the fluorescence changes due to oxidation by H_2_O_2_, comparing these parameters to those of HyPer3. Purified HyPer3 protein was pH-sensitive in the pH range of 5.5–8.5, such that for any pH value in this range, its maximal fluorescence change at a shift of +/−0.5 pH units was comparable or exceeded the fluorescence change due to the reaction of HyPer3 with 200 µM H_2_O_2_ by up to 6-fold (Table 1). The NeonOxIrr indicator, within the same pH range of 5.5–8.5, demonstrated lower sensitivity to pH changes at +/−0.5 units; however, such pH changes impacted the dynamic range of NeonOxIrr by 6%–52%. We defined the pH range of 7.0–8.5 as the “working” range of NeonOxIrr, where fluorescence changes due to a pH shift of +/−0.5 units were 5-fold less than those due to the reaction with saturating amounts of H_2_O_2_. At physiological pH 7.5 characteristic of the cytoplasm, NeonOxIrr was 4-/20-fold less pH-sensitive at alkalization/acidification at +/−0.5 units than HyPer3. Thus, the NeonOxIrr indicator significantly outperformed the standard HyPer3 indicator in terms of pH sensitivity in both the reduced and oxidized states and is therefore suitable for detection of H_2_O_2_ under conditions with varying pH.

We further compared the chromophore maturation, photostability and oligomeric state of the NeonOxIrr indicator to the corresponding parameters of HyPer3 and mEGFP. At 37 °C, NeonOxIrr exhibited 6- and 3-fold faster maturation rates than HyPer3 and mEGFP, respectively (Figure 1e). Under wide-field illumination at the 470/40 nm wavelength, NeonOxIrr photobleached 2.1-fold faster than HyPer3 and similarly to mEGFP (Figure 1f). The purified NeonOxIrr and HyPer3 indicators migrated on a semi-native gel as weak dimers in contrast to mEGFP, demonstrating their mainly monomeric state under the same conditions (ESI Appendix A).

We next assessed the contrast, sensitivity and reduction half-times of NeonOxIrr and HyPer3 indicators in bacterial suspensions. The maximally achievable fluorescence contrast of the NeonOxIrr indicator in bacterial suspensions at a 100–200 µM concentration of external H_2_O_2_ was similar to that of control HyPer3 (Table 1, ESI Appendix A). At a lower concentration of external H_2_O_2_ (1–20 µM), NeonOxIrr demonstrated a 1.8–2.7-fold higher ΔF/F response than HyPer3. NeonOxIrr was characterized by a reduction half-time of 280 min in bacteria, which was 4.6-fold longer than that for HyPer3 (ESI Appendix A). Hence, in bacterial suspensions, NeonOxIrr revealed similar fluorescence contrast at saturating H_2_O_2_ concentrations, higher apparent sensitivity to lower H_2_O_2_ concentrations and a significantly slower reduction rate compared to HyPer3.

We further compared the rates of NeonOxIrr and HyPer3 oxidation with atmospheric oxygen and external H_2_O_2_. In the presence of catalase, which degraded residual amounts of H_2_O_2_, and at r.t. (25 °C), purified redox indicators were gradually oxidized by oxygen at rates of 3.4 and 0.19 M^−1^s^−1^, respectively (ESI Appendix A); however, in these conditions the probes may have been oxidized by extremely low amounts of H_2_O_2_ generated by traces of metals in the reaction mixture [22]. The addition of H_2_O_2_ dramatically increased oxidation rate constants for NeonOxIrr and HyPer3 to values of 16,000 and 780 M^−1^s^−1^, or by 4706- and 4105-fold, respectively. A temperature increase by 12 degrees to 37 °C further accelerated the H_2_O_2_ oxidation rates of the NeonOxIrr and HyPer3 indicators to values of 22,100 and 1443 M^−1^s^−1^ or by 1.4- and 1.9-fold, respectively. According to H_2_O_2_ oxidation rate constants, NeonOxIrr reacted with H_2_O_2_ by 20.5– and 15.3-fold faster than HyPer3 at 25 °C and 37 °C, respectively. A faster reactivity of NeonOxIrr coincided with its higher apparent sensitivity to low 1–20 µM concentrations of H_2_O_2_ compared with HyPer3. Hence, the NeonOxIrr indicator reacted with H_2_O_2_ specifically and 15–20-fold faster than HyPer3.

### 2.3. Response of NeonOxIrr to external H_2_O_2_ in Different Organelles of Mammalian Cells

To characterize and compare the properties of NeonOxIrr and HyPer3 indicators in cultured mammalian cells, we expressed them in the cytoplasm of HEK293T mammalian cells and assessed the sensitivity and dynamics of their response to the external H_2_O_2_. After the addition of a saturating 200 µM concentration of H_2_O_2_, the indicators reached their maximal fluorescence change in 3–5 min and gradually reduced to original fluorescence values afterwards (Figure 2a). According to the oxidation rate constants, NeonOxIrr oxidized 1.7-fold faster than control HyPer3 (ESI Appendix A and ESI Appendix A). Notably, the oxidation rate for the purified NeonOxIrr protein was 117-fold higher compared to that in the cytoplasm of mammalian cells. This difference could be explained by the existence of a concentration gradient of peroxide across the plasma membrane [23,24] as well as the impact of rate-limiting steps other than indicator oxidation, such as the diffusion of H_2_O_2_ through the cellular membrane. According to reduction half-time values, the NeonOxIrr indicator was reduced in the cytoplasm of HEK293T cells 7.6-fold slower than HyPer3 (ESI Appendix A). 

The maximal fluorescence dynamic range of NeonOxIrr observed upon the addition of 200 µM H_2_O_2_ to HEK 293T cells was 2.3-fold or 130%, while HyPer3 had a dynamic range of 2.8-fold or 180% (Figure 2b and ESI Appendix A). The ΔF/F response of the NeonOxIrr and HyPer3 indicators at reduced external H_2_O_2_ concentrations of 20 and 50 µM dramatically dropped down to 27%, 40% and 25%, 50%, respectively. NeonOxIrr was able to detect an external H_2_O_2_ concentration as low as 5 µM with a ΔF/F response of 11%. The brightness of NeonOxIrr in mammalian cells was approximately 10-fold higher than that for HyPer3. 

We next related the response of NeonOxIrr indicator to commonly used R-GECO1 calcium indicator and demonstrated simultaneous two-color imaging of H_2_O_2_ and calcium transients in mammalian cells. We transiently co-expressed NeonOxIrr hydrogen peroxide indicator and R-GECO1 calcium indicator in HeLa cells. The response of NeonOxIrr to 200 µM H_2_O_2_ was significantly 3-fold lower as compared with ionomycin-induced calcium response for the commonly used R-GECO1 calcium indicator (Figure 2d, ESI Appendix A, and ESI Appendix A). Therefore, in combination with R-GECO1 calcium indicator, NeonOxIrr indicator allows simultaneous monitoring of H_2_O_2_ and calcium transients in mammalian cells using two-color confocal microscopy.

Because of high brightness and monomeric state of NeonOxIrr indicator we further assessed localization and response of NeonOxIrr indicator in N-terminal fusion with intermediate filament protein, vimentin. Both vimentin-NeonOxIrr and its control vimentin-NeonOxIrr/C199S fusions were properly localized in the intermediate filaments of HeLa Kyoto cells (ESI Appendix A). The expression levels of the NeonOxIrr alone and vimentin-NeonOxIrr fusion estimated relative to co-expressed mCherry protein were the same (ESI Appendix A). Vimentin-NeonOxIrr fusion demonstrated high 137±33 % ΔF/F response to the 200 µM saturating amounts of externally added H_2_O_2_ (ESI Appendix A) in opposite to vimentin-NeonOxIrr/C199S fusion which did not react with external H_2_O_2_ (ESI Appendix A). It meant that NeonOxIrr indicator was reduced under these conditions. Therefore, NeonOxIrr indicator in fusion with vimentin protein demonstrated proper localization and preserved high ΔF/F response of 137%.

Hence, when expressed in the cytoplasm of mammalian cells, NeonOxIrr demonstrated a 10-fold higher brightness, 1.4-fold lower ΔF/F response, and 7.6-fold longer reduction time compared to control HyPer3 hydrogen peroxide indicator and 3-fold lower response compared to established R-GECO1 calcium indicator.

In addition to the cytoplasm, we characterized the localization and response of the NeonOxIrr indicator to a 200 µM concentration of external H_2_O_2_ in various organelles of HEK293T cells, e.g., in the matrix and intermembrane space (IMS) of mitochondria, in the nucleus, on the plasma membrane, in peroxisomes, in the Golgi complex, and in the endoplasmic reticulum (ER). In all tested organelles, the NeonOxIrr indicator was properly localized (Figure 3). In the nucleus, as well as in the matrix and IMS of mitochondria, NeonOxIrr reacted with H_2_O_2_ at a 2.45-, 2.44-, and 2.27-fold signal increase, respectively, which was similar to the values in the cytosol (Figure 2c, ESI Appendix A). NeonOxIrr revealed a 49% response in peroxisomes, which was 2.7-fold lower than its response in the cytosol. The responses of the NeonOxIrr indicator were further reduced to 28%, 21%, and 17% when targeted for the cytoplasmic side of the membrane, ER lumen, and Golgi complex, respectively; compared with cytosolic response, the responses were 4.6-, 6.2-, and 7.6-fold lower, respectively. 

To confirm the specificity of the reaction of NeonOxIrr with H_2_O_2_ and to control for variations of parameters such as pH, we introduced a Cys199Ser mutation that was expected to block the reaction of the OxyR part of the indicator with H_2_O_2_ (which normally results in the formation of the Cys199–Cys208 bond). Notably, NeonOxIrr carrying the Cys199Ser mutation did not respond to the addition of 200 µM H_2_O_2_ when expressed in the cytoplasm of HeLa Kyoto cells, indicating the specificity of the indicator’s response (ESI Appendix A).

### 2.4. Response of NeonOxIrr to Endogenously Produced ROS in the Cytoplasm, on Plasma Membrane and in Mitochondria of Mammalian Cells

After demonstrating the specificity and efficiency of the NeonOxIrr response to exogenously added H_2_O_2_ in vitro and in cultured cells, we sought to assess the indicator’s ability to sense the production of endogenous H_2_O_2_ in live cells. We examined the response of the active and inactive (carrying C199S mutation) versions of the indicator to several stimuli reported to elicit endogenous production of H_2_O_2_ production in cells [5,6].

Thapsigargin inhibits Ca^2+^-ATPases in the sarco-endoplasmic reticulum [25] and induces response of the HyPer indicators. When 10 µM thapsigargin was added to the HeLa cells co-expressing the green Mito-NeonOxIrr redox indicator (or its C199S mutant) in the matrix of mitochondria and red R-GECO1 calcium indicator in the cytoplasm, the anticipated increase in Ca^2+^ concentration was detected (ESI Appendix A); however, we could not detect a reliable difference in the responses of Mito-NeonOxIrr (ESI Appendix A) and its Mito-NeonOxIrr/C199S mutant (ESI Appendix A) in the mitochondria matrix, when normalized to the maximal Mito-NeonOxIrr fluorescence achieved after the H_2_O_2_ addition. 

Similarly, addition of 100 ng/mL epidermal growth factor (EGF) to the transfected HeLa cells expressing the indicators in cytoplasm did not affect the response of the NeonOxIrr indicator (ESI Appendix A) or its NeonOxIrr/C199S mutant (ESI Appendix A), when normalized to the maximal NeonOxIrr fluorescence achieved after the subsequent exogenous H_2_O_2_ addition. The response of the vimentin-NeonOxIrr fusion protein to addition of EGF stimulus (ESI Appendix A) was similar to that for the control vimentin-NeonOxIrr/C199S fusion (ESI Appendix A). We confirmed that EGF was functional in the HeLa cells by observing EGFR-GFP endocytosis, as revealed by the formation of green dots (ESI Appendix A) [26]. 

We next examined the action of apoptosis-inducing drug staurosporine [27] in HeLa cells transiently expressing Mito-NeonOxIrr indicator or its C199S mutant version in the matrix of mitochondria. Addition of 1 µM of staurosporine resulted in a dramatic increase of NeonOxIrr fluorescence in the mitochondria within 8.5–10 min, starting several hours after staurosporine addition and varying for individual cells (Figure 4a). Such increase in the green fluorescence was not observed in cells expressing Mito-NeonOxIrr/C199S mutant which lacks sensitivity to H_2_O_2_ (Figure 4b). 

To address low sensitivity of NeonOxIrr to endogenously produced ROS in cytosol of the cells, we reasoned that targeting of NeonOxIrr closer to the sites of endogenous ROS production might increase local concentration of ROS and improve sensitivity of NeonOxIrr. The known site of H_2_O_2_ or O_2_^−^ production in mammalian cells is the plasma membrane [28]. NAD(P)H oxidases (NOXs) are activated by various growth factors and cytokines and generate H_2_O_2_ or O_2_^−^ for signaling processes such as proliferation, migration, and survival [29]. To examine possibility of H_2_O_2_ sensing by NeonOxIrr at the plasma membrane we constructed the N-terminal fusion proteins of NeonOxIrr and its C199S mutant with Lyn-peptide that targeted the fusion proteins to the plasma membrane, expressed them in HeLa cells and studied their response to external H_2_O_2_ and EGF-induced endogenous ROS (ESI Appendix A). As compared with cytosol, 79% of Lyn-NeonOxIrr was oxidized when localized in the plasma membrane because it demonstrated 28% vs. 132% ΔF/F response to the external H_2_O_2_ on the plasma membrane and cytosol, respectively (Figure 2c and ESI Appendix A). The control Lyn-NeonOxIrr/C199S did not responded to the external H_2_O_2_ (ESI Appendix A). We did not notice difference in the responses of Lyn-NeonOxIrr (ESI Appendix A) and control Lyn-NeonOxIrr/C199S (ESI Appendix A) to EGF addition. The failure to detect ROS on plasma membrane with NeonOxIrr might be related with its mostly oxidized state. We reasoned that utilization of the hydrogen peroxide indicator with reduced sensitivity to H_2_O_2_ might resolve this issue. With this suggestion we fused our unpublished H_2_O_2_ indicator, called NeonOxE, which has 11-fold lower sensitivity to H_2_O_2_ (oxidation rate 1400 ± 50 M^−1^s^−1^ vs. 16000 ± 1000 M^−1^s^−1^ for NeonOxE and NeonOxIrr, respectively) to the Lyn-peptide, expressed it on the plasma membrane of HeLa cells and measured its response to external H_2_O_2_ and EGF-induced endogenous ROS. As compared to cytosol, 73% of the NeonOxE indicator was oxidized at the plasma membrane and demonstrated high 119% ΔF/F response to the addition of 200 µM saturating concentration of H_2_O_2_ (ESI Appendix A). Membrane-localized response for NeonOxE was 4.3-fold higher as compared with the same response for NeonOxIrr. We detected very weak and barely visible positive response of NeonOxE indicator to the addition of EGF (ESI Appendix A). The control Lyn-NeonOxE/C199S fusion protein did not respond to both the external 200 µM H_2_O_2_ and the addition of EGF (ESI Appendix A). Comparison of ΔF/F values for Lyn-NeonOxE and Lyn-NeonOxE/C199S fusion proteins in HeLa cells after addition of EGF showed statistically significant difference (ESI Appendix A). We suggested that to get more pronounced response of the H_2_O_2_ indicator to the growth factor addition such as EGF on the plasma membrane we need to use H_2_O_2_ indicator with minimal extent of oxidation on the plasma membrane. 

Thus, the NeonOxIrr indicator did not detect endogenous production of ROS in the cytoplasm, plasma membrane or matrix of the mitochondria of HeLa cells after addition of thapsigargin or EGF, but was able to robustly and specifically detect the production of endogenous ROS in the matrix of mitochondria in HeLa cells undergoing staurosporine-induced apoptosis.

### 2.5. Visualization of Optogenetically Generated ROS in Mammalian Cells

Illumination of KillerRed red fluorescent protein with yellow light results in the generation of active forms of oxygen or ROS which are primarily represented by superoxide and singlet oxygen [30]. We sought to determine whether our redox indicators can sense such optogenetically generated ROS in mammalian cells. Illumination with yellow light (for 10 or 100 s) of HEK293T cells expressing H2B-HyPer3 fusion in the nucleus did not cause noticeable fluorescence changes for HyPer3, as compared with the reaction of HyPer3 to the addition of 200 µM H_2_O_2_ (ESI Appendix A). Furthermore, the co-expression of KillerRed with HyPer3 in the nucleus did not cause notable fluorescence changes for HyPer3 after illumination with yellow light (for 10 or 100 s), as compared with its response to 200 µM H_2_O_2_ (ESI Appendix A). However, after optogenetic ROS generation, HyPer3 did not respond to the addition of H_2_O_2_ (ESI Appendix A). We also observed that after Hyper3 oxidation with H_2_O_2_, the optogenetic production of ROS completely quenched the fluorescence of maximally oxidized HyPer3 (ESI Appendix A). Thus, HyPer3 indicator in the reduced and oxidized states irreversibly reacted to optogenetically produced ROS and this process was accompanied by no change or quenching of its fluorescence, respectively; this reaction subsequently suppressed the ability of HyPer3 to sense H_2_O_2_.

In contrast, the NeonOxIrr indicator responded to the optogenetic production of ROS with an increase in its green fluorescence (Figure 5). The amplitude of the NeonOxIrr response to optogenetic ROS was comparable to the NeonOxIrr response to 200 µM H_2_O_2_. The optogenetically induced fluorescence of NeonOxIrr underwent decay at a diminished rate compared to the H_2_O_2_-induced response. Thus, unlike HyPer3, NeonOxIrr showed a significant fluorescence response to optogenetically produced ROS in live mammalian cells.

### 2.6. Engineering of the Ratiometric Version of the NeonOxIrr Peroxide Indicator

For quantitative reporting of H_2_O_2_ production, we engineered a ratiometric version of the NeonOxIrr indicator and validated its performance in bacteria. To enable quantitative measurements, we generated several fusions of the NeonOxIrr indicator with the red fluorescent protein mCherry and tested them for the maturation and response to H_2_O_2_ in bacteria. These constructs varied regarding the order of the NeonOxIrr and mCherry genes and the linkers P2A, L3, and L2 with a length of 42, 21, and 16 amino acids, respectively (ESI Appendix A). We chose self-cleavable P2A peptide because it is suggested to be cleaved after the expression of the fusion protein and because the expression of two proteins connected via P2A peptide is stoichiometric in mammalian cells [31]. L3 and L2 linkers were chosen because they had shorter lengths vs. P2A peptide.

When expressed in bacteria, the fluorescence of the generated fusion proteins in green and red channels demonstrated dependence on the temperature of expression, the concentration of the arabinose inductor, the order of the proteins in the fusion, and the length of the linker between the proteins (ESI Appendix A). The NeonOxIrr-P2A-mCherry fusion showed the greatest response to external H_2_O_2_ (ESI Appendix A and ESI Appendix A). The sensitivity of NeonOxIrr in the fusion with P2A-mCherry was similar to that of NeonOxIrr expressed alone, both constructs reaching maximal contrast at 4 µM of H_2_O_2_ (ESI Appendix A). Thus, the NeonOxIrr-P2A-mCherry fusion showed optimal performance in bacteria compared to other screened fusions and had sensitivity to H_2_O_2_ that was similar to that of NeonOxIrr.

### 2.7. Optimization of Chemical Fixation Protocol for Bacteria Expressing the Ratiometric NeonOxIrr-P2A-mCherry Indicator

To demonstrate the possibility of ex vivo H_2_O_2_ visualization with the NeonOxIrr-P2A-mCherry ratiometric indicator, we developed an alkylation-fixation protocol using IAM and NEM alkylating reagents. With this aim, we studied the influence of various factors, such as the concentration of alkylating reagents, pH, temperature, treatment with H_2_O_2_, and the concentration of paraformaldehyde (PFA) on the fluorescence dynamic range of the NeonOxIrr-P2A-mCherry indicator after the alkylation-fixation procedure (ESI Appendix A and ESI Appendix A). Both of the tested alkylating reagents, NEM and IAM, allowed maintaining a high dynamic range (2.5- and 1.6-fold, respectively) for the NeonOxIrr-P2A-mCherry response to H_2_O_2_ after bacterial cells fixation. The most optimal conditions for alkylation and fixation were selected based on the highest green-to-red signal ratio in response to H_2_O_2_ vs. vehicle ex vivo. In the case of NEM, alkylation at pH 8.5 at concentrations of 0.8 mM or 4 mM at 25 °C resulted in the highest ex vivo signal change of 2.5-fold. In the case of IAM, alkylation at pH 7.4 at a concentration of 0.2 mM and at 4–25 °C provided a 1.6-fold maximal ex vivo signal change. The blocking of endogenous peroxidase activity by treatment with 0.3% H_2_O_2_ was incompatible with the developed alkylation-fixation protocol of the NeonOxIrr-P2A-mCherry peroxide indicator, resulting in the loss of the ex vivo signal (ESI Appendix A). The fixation of the fusion indicator after alkylation can be efficiently carried out with 1%–4% PFA (ESI Appendix A). Thus, we found optimal alkylation-fixation conditions with 0.8–4 mM NEM at 25 °C, pH 8.5, which preserved the high 2.5-fold fluorescence contrast of the NeonOxIrr-P2A-mCherry indicator in response to H_2_O_2_ ex vivo in bacterial cells.

### 2.8. Response of the Ratiometric Versions of the NeonOxIrr Indicator to External H_2_O_2_ in Mammalian Cells

We next tested the response of the fusions of the NeonOxIrr indicator with mCherry to external H_2_O_2_ in mammalian cells. The six generated fusions described above were re-cloned into the pLU-CMV mammalian vector and transfected into HeLa Kyoto cells. We observed the reaction of the NeonOxIrr-P2A-mCherry to 200 μM H_2_O_2_ in HeLa cells under a confocal microscope (Figure 6a). As expected, the administration of H_2_O_2_ caused an increase in fluorescence only in the green channel, with no changes observed in the red channel (Figure 6a). At a saturating concentrations of H_2_O_2_ (200 μM), all six indicators exhibited the same fluorescence increase of 2.3–2.7-fold, with the exception of the NeonOxIrr-L2-mCherry fusion with a short linker which showed a 4.7-fold lower ΔF/F response than the other tested constructs (ESI Appendix A). Thus, similar to the original non-fused NeonOxIrr indicator, most ratiometric versions of NeonOxIrr demonstrated 2.3–2.7-fold fluorescence contrasts in response to the exogenous H_2_O_2_ in HeLa cells.

### 2.9. Ex vivo H_2_O_2_ Detection in HeLa Cells Expressing Ratiometric NeonOxIrr-P2A-mCherry

The alkylation-fixation protocol based on the NEM treatment that was found to be optimal in bacterial cells was next validated in mammalian cells. For alkylation, we examined a 1 mM concentration of NEM that was optimal for bacteria in our experiments, as well as a 20 mM concentration used in other studies [7,8,9]. After oxidizing the indicator with 200 μM of H_2_O_2_ (a saturating amount), HeLa cells were alkylated and fixed with 4% PFA. We then studied how the contrast of the fusion indicator changes over time during 12 days of storage at 4 °C after fixation (Figure 6b and ESI Appendix A). As a negative control, we used an inactive mutant indicator NeonOxIrr/C199S-P2A-mCherry (ESI Appendix A). Immediately after alkylation with 1 mM NEM and fixation with 4% PFA, we observed a 2.5-fold fluorescence green-to-red ratio increase for the NeonOxIrr-P2A-mCherry indicator ex vivo (ESI Appendix A). The indicator ratiometric signal dropped to 1.7-fold during storage for 3 days after fixation and was approximately at the same level up to 12 days after fixation (Figure 6b and ESI Appendix A). At a 20 mM NEM concentration, the indicator’s ex vivo ratiometric signal was lower; this may be due to the fact that at a higher concentration of the alkylating reagent, more cysteine residues may be alkylated, which can affect the conformation of the protein (ESI Appendix A). In the case of the NeonOxIrr/C199S-P2A-mCherry control mutant, its fluorescence green-to-red ratio increased 1.32-fold immediately after alkylation-fixation (ESI Appendix A), and on the 3rd and 12th days of storage, it decreased to 1.13- and 1.06-fold, respectively (ESI Appendix A).

Thus, the alkylation-fixation procedure with 1 mM NEM and 4% PFA was effective for HeLa cells expressing the NeonOxIrr-P2A-mCherry indicator, resulting in high 2.5- and 1.7-fold dynamic range ex vivo on the first and twelfth days after alkylation-fixation, respectively.

### 2.10. In vivo Response of Ratiometric NeonOxIrr-P2A-mCherry Indicator to External H_2_O_2_ in Living Dissociated Neuronal Cultures

Because the alkylation-fixation protocol did not reveal H_2_O_2_ signal ex vivo for either neuronal cultures (ESI Appendix A, Figure 6c, and ESI Appendix A) or the mouse cortex (ESI Appendix A, ESI Appendix A), we reasoned that the NeonOxIrr-P2A-mCherry indicator may be oxidized in neurons by endogenous ROS. To examine this possibility, we expressed NeonOxIrr-P2A-mCherry indicator in the neuronal cultures and studied its response to the external H_2_O_2_ in live neurons using confocal time-lapse imaging. Neuronal cultures were transduced with the rAAV particles carrying NeonOxIrr-P2A-mCherry under the control of the CAG promoter. On day 10–14 after transduction, we observed the expression of the green NeonOxIrr indicator and the red mCherry fluorescent protein with localization in the cytoplasm and nucleus (Figure 6d). When 200 μM H_2_O_2_ was added, the NeonOxIrr green fluorescence increased by 3.2-fold within 3 min, while the red fluorescence of mCherry remained unchanged (Figure 6d). Thus, the dynamic range of NeonOxIrr-P2A-mCherry in response to external H_2_O_2_ in cultured neurons was highly similar to that in HeLa cells. 

### 2.11. Development of the Alkylation-Fixation Protocol with NEM and Various Detergents for H_2_O_2_ Detection in Neuronal Cultures Expressing Ratiometric NeonOxIrr-P2A-mCherry

As the NeonOxIrr-P2A-mCherry indicator demonstrated a high fluorescence response to H_2_O_2_ in living cultured neurons, we asked whether the alkylation-fixation protocol did not work in neurons because of problems with NEM penetration across the plasma membrane of neurons in the dissociated neuronal culture as well as in the mouse brain. Indeed, the composition of the neuronal membrane differs from that of other animal cells due to the presence of various channels, proteins, lipids, etc. [32]. To improve NEM penetration across the neuronal plasma membrane, we performed alkylation of neuronal cultures expressing NeonOxIrr-P2A-mCherry with 1 mM NEM in the presence of different mild non-ionic detergents, such as saponin, Tween−20, and NP−40, which facilitate permeabilization of the plasma membrane [33]. In addition, we increased the NEM concentration to 10 mM. The increased NEM concentration as well as the addition of saponin or Tween−20 did not result in a positive ex vivo signal change between the H_2_O_2_-treated and non-treated neuronal cultures expressing NeonOxIrr-P2A-mCherry (Figure 7a). However, the presence of 0.01% NP−40 detergent led to a 1.5-fold fluorescence ex vivo dynamic range (Figure 7a, green bars). Hence, the presence of the non-ionic detergent NP-40 may have facilitated the NEM permeability of the neuronal plasma membrane and showed a 1.5-fold ex vivo dynamic range of the response to H_2_O_2_ for the NeonOxIrr-P2A-mCherry indicator on neuronal cultures. Therefore, we used NP-40 as the detergent of choice for the ex vivo visualization of H_2_O_2_ in the mouse brain for all subsequent alkylation-fixation experiments.

### 2.12. Ex vivo H_2_O_2_ Detection in the Mouse brain Expressing Ratiometric NeonOxIrr-P2A-mCherry using a Modified Fixation Protocol

As the modified alkylation-fixation protocol with NEM and NP−40 detergent allowed ex vivo visualization of H_2_O_2_ in neuronal cultures expressing NeonOxIrr-P2A-mCherry indicator, we applied it to the mouse brain. We infected both hemispheres of the mouse cerebral cortex with rAAV particles carrying the NeonOxIrr-P2A-mCherry. One week after infection, we injected H_2_O_2_ into the left and buffer with no H_2_O_2_ into the right brain hemispheres to the areas expressing the NeonOxIrr-P2A-mCherry indicator. We probed several conditions for alkylation-fixation. First, we used the concentrations of reagents for oxidation–alkylation in the presence of 0.01% NP−40 that were optimal for the neuronal cultures. There was a non-significant 1.25-fold (*p* = 0.0512) difference in the green-to-red fluorescence ratios between hemispheres with or without 200 µM external H_2_O_2_ (Figure 7b, red bars). Second, to improve NEM penetration into neurons, we increased the concentrations of NEM and NP−40 by 5-fold. However, this change resulted in a non-significant (*p* = 0.1798) decrease in the green-to-red fluorescence ratio after treatment with the external 200 µM H_2_O_2_ (Figure 7b, green bars). Finally, we asked whether H_2_O_2_ concentration in the brain parenchyma may be diluted upon injection. When the concentration of the injected H_2_O_2_ was increased to 2 mM, we observed a significant 1.50-fold (*p* = 0.0036) difference in the green-to-red fluorescence ratio between the H_2_O_2_-treated and buffer-treated hemispheres (Figure 7b, blue bars). 

We also performed immunohistochemical staining of fixed slices on markers of astrocytes (GFAP) and neuronal cells (NeuN) (for more details, see ESI Appendix A). We found that the alkylation-fixation procedure did not affect the epitopes of GFAP or NeuN, or the specificity of antibody binding in brain slices (ESI Appendix A and ESI Appendix A). 

Thus, we were able to visualize external H_2_O_2_ in the mouse brain cortex ex vivo with 1.5-fold dynamic range after alkylation-fixation using the ratiometric NeonOxIrr-P2A-mCherry indicator. 

### 2.13. Ex vivo Response of NeonOxIrr-P2A-mCherry to Endogenously Produced ROS in the Brain after γ-Irradiation.

Several studies reported that the exposure of mammalian cells to ionizing radiation (IR) increased the production of endogenous ROS in the irradiated cells [34,35,36]. The production of ROS in the brain as early as 1 h after exposure to 5 Gy of γ-rays was proposed based on the appearance of mtDNA fragments in the cytosolic fractions of the brain [37]. Furthermore, irradiation of neuronal stem cell cultures with a low 5 Gray dose of γ-rays leads to an increase in the number of active forms of superoxide with a maximum at 12 h, coinciding with the maximum level of cell apoptosis [31,38]. Increased H_2_O_2_ production also occurs during apoptosis [25]. The H_2_O_2_ production in the cytoplasm or mitochondria was shown in the case of induced apoptosis of HeLa cells by the Apo2L/TRAIL protein [5] or staurosporine (Figure 4) using the HyPer and NeonOxIrr genetically encoded redox indicators, respectively.

We sought to test the NeonOxIrr-P2A-mCherry indicator in combination with the developed alkylation-fixation protocol to detect ROS production in the cortex and hippocampus of the mouse brain after 5 Gy γ-irradiation. We delivered rAAV particles carrying CAG-NeonOxIrr-P2A-mCherry construct into the cortex and hippocampus of mice. One week after infection, the mice were irradiated with 5 Gy γ-irradiation or were sham-treated without irradiation. At 35 min or 16 h after irradiation, the mice expressing NeonOxIrr-P2A-mCherry in the cortex or hippocampus were perfused with 1 mM NEM, 0.01% NP−40 for 10 min followed by fixation with 4% PFA. The brain slices were then obtained and imaged using confocal fluorescence microscopy. At 35 min after 5 Gy γ-irradiation, the NeonOxIrr-P2A-mCherry green-to-red fluorescence ratio in the mouse cortex decreased significantly by 1.37-fold (*p* = 0.0046; ESI Appendix A). At 16 h after 5 Gy γ-irradiation, the NeonOxIrr-P2A-mCherry green-to-red fluorescence ratio in the mice hippocampus was non-significantly 1.29-fold (*p* = 0.4) higher than the analogous ratios for the non-irradiated control group (Figure 7c). Thus, we were not able to detect ROS production in the mouse brain 0.5 or 16 h after 5 Gy γ-irradiation using the NeonOxIrr-P2A-mCherry indicator and alkylation-fixation protocol.

## 3. Discussion

Using directed molecular evolution in bacteria, we generated a new genetically encoded green hydrogen peroxide indicator NeonOxIrr. Compared to the HyPer family indicators, a current standard in the field, NeonOxIrr indicator has significantly higher brightness, contrast, and sensitivity, and has significantly slower reduction kinetics after reacting with H_2_O_2_ in living cells. NeonOxIrr allowed visualization of the external H_2_O_2_ in living cultured HEK293T, HeLa and neuronal cells. It also demonstrated expected localization and functionality in the organelles of mammalian cells. Furthermore, we developed a protocol of NeonOxIrr alkylation-fixation with NEM which enabled the ex vivo visualization of external H_2_O_2_ in HeLa cells; modified version of protocol with NEM and NP−40 worked well for the cultured neurons and in the mouse brain cortex (ESI Appendix A). 

The slowly reducible phenotype of the NeonOxIrr indicator opens several possibilities that are not fully accessible with available fast-reducible redox indicators. First, NeonOxIrr may be used as an integrator of the flux of redox signals in vivo over a particular period of time. Second, it may be able to detect particularly weak signals if they sum up within such time period. Third, the indicator’s slow reduction phenotype provides sufficient time for an optimized alkylation-fixation reaction in order to preserve indicator’s fluorescence response to H_2_O_2_ for ex vivo imaging. 

The fluorescent part of the NeonOxIrr indicator is based on a novel circularly permutated version of the mNeonGreen fluorescent protein, called cpmNeonGreen. Importantly, the permutated version preserved the high brightness and pH stability of the parental mNeonGreen protein. Therefore, cpmNeonGreen may be a promising platform for the development of bright and pH-stable genetically encoded indicators for other signals, e.g., calcium indicators. We recently reported a NTnC calcium indicator based on mNeonGreen with insertion of the troponin C domain [39]. Compared with mNeonGreen, the insertion version of cpmNeonGreen requires less effort to replace the fluorescent moiety in the available indicators, most of which are based on circularly permutated proteins such as cpGFP, cpEYFP, and cpmApple.

Yet another important feature of the NeonOxIrr indicator is its high pH stability, which may become critical in some contexts and may help to avoid potential artifacts in detection of ROS in live cells that are due to pH changes. For instance, physiological pH changes in mitochondria can be so dramatic as to hinder the measurements of redox status using conventional mitochondria-targeted sensors [21]. Although the NeonOxIrr indicator is stable within a wide range of pH and has reliably reported production of ROS in the matrix of mitochondria (Figure 4), we strongly recommend comparing the results obtained with NeonOxIrr and with its NeonOxIrr/C199S version which lacks sensitivity to H_2_O_2_, to be used as a control for pH-changes, cellular volume changes and other conditions.

The NeonOxIrr indicator may react not only with H_2_O_2_ but also with other oxidants. The purified NeonOxIrr indicator was non-specifically oxidized by atmospheric oxygen at a 4706-fold lower rate compared to that of H_2_O_2_ (ESI Appendix A). However, even at that low rate, 40 min was sufficient for its complete oxidation; this should be taken into account when interpreting the long-term changes in the florescence of the NeonOxIrr indicator in living cells. At the same time, we cannot exclude the possibility that under our experimental conditions the slow oxidation of the probe was due to low amounts of H_2_O_2_ generated in the reaction mixture by redox cycling between oxygen, thiols, and trace amounts of metals. Also note that in the experiments with KillerRed fluorescent protein in mammalian cells (Figure 5) NeonOxIrr indicator may have also reacted, besides H_2_O_2_, with superoxide and/or singlet oxygen, which can be optogenetically generated by KillerRed upon illumination with yellow light; in that case NeonOxIrr may be considered, to some extent, as a ROS, rather than a purely H_2_O_2_ indicator. 

When interpreting the NeonOxIrr response in cell organelles, an important consideration is that at saturating H_2_O_2_ concentrations the response was maximal in the cytosol, nuclei, lumen and IMS of mitochondria of the HEK293T cells, but was significantly reduced by 2.7-, 6.0-, 6.2-, and 7.6-fold in the peroxisomes, cytoplasmic side of the membrane, ER, and Golgi, respectively (Figure 2c). A reduced response of the NeonOxIrr indicator correlates with a more oxidative redox state in ER compared to cytosol (~30-fold decreased GSH/GSSG ratio in ER compared with cytosol) [40]. A redox state may also affect the response of the NeonOxIrr indicator in Golgi, cytoplasmic membrane side and peroxisomes. The oxidation state of Hyper in different organelles of HEK 293 cells was previously examined using a treatment of cells expressing HyPer with 1 mM DTT or 50 µM H_2_O_2_ to generate fully reduced and fully oxidized conditions, respectively. In the cytosol, mitochondria, mitochondrial IMS and peroxisomes, 10%–16% of HyPer was in the oxidized form, whereas ~70% was in the oxidized form in the ER. In accordance with these data, both the Hyper and NeonOxIrr indicators showed a large dynamic range of the reaction with H_2_O_2_ when localized in the cytosol, mitochondria, and mitochondrial IMS and a significantly lower response at the ER localization. Both indicators also demonstrated a large response to H_2_O_2_ treatment in the nucleus. In peroxisomes, NeonOxIrr, unlike Hyper, demonstrated a diminished contrast in response to H_2_O_2_. The low contrast in response to H_2_O_2_ treatment of the NeonOxIrr indicator localized in the Golgi and plasma membrane was not tested for Hyper. A decreased response of NeonOxIrr in these cellular compartments is likely due to a more oxidative redox state or a higher peroxide resting concentration.

The generated ratiometric version of the NeonOxIrr-P2A-mCherry indicator showed an even distribution of mCherry protein in HeLa cells, but uneven puncta-like distribution of mCherry in neurons (Figure 6a vs. 6d and ESI Appendix A). The fluorescence intensity of mCherry in these puncta-like structures was significantly higher than in the surrounding areas, and during imaging in neurons, an appropriate exposure time should be applied to avoid overexposure. The mCherry puncta may correspond to lysosomes [41]; therefore autophagy may affect the number of these puncta [42] and influence the green-to-red ratio of the NeonOxIrr-P2A-mCherry indicator. If deemed critical, the problem may be resolved by utilizing other than mCherry red fluorescent proteins such as FusionRed [41].

The low sensitivity to intracellular ROS may limit in vivo applications of the NeonOxIrr indicator. The purified NeonOxIrr and the NeonOxIrr expressed in mammalian cells demonstrated a faster oxidation rate with external H_2_O_2_ than HyPer3. Compared with HyPer3, NeonOxIrr revealed higher and similar sensitivity to low 1–20 µM external H_2_O_2_ concentrations when expressed in bacteria and mammalian cells, respectively. However, using NeonOxIrr, we could not detect endogenous ROS production in the mitochondria or cytoplasm of the HeLa cells stimulated with thapsigargin or EGF, respectively (ESI Appendix A and ESI Appendix A, ESI Appendix A). This result was unanticipated because indicators of the HyPer family robustly revealed intracellular H_2_O_2_ in similar conditions of stimulation [5,6]. According to these criteria, NeonOxIrr will require further optimization to match the sensitivity of HyPers to endogenous H_2_O_2_.

NeonOxIrr has significantly slower reduction kinetics in bacteria, HEK293T, HeLa cells, and neurons than other H_2_O_2_ indicators, this feature enabling ex vivo detection of H_2_O_2_ via the alkylation-fixation procedure. Reversible peroxide indicators with faster reduction kinetics, such as indicators of the HyPer family [5,6], may become partially reduced after H_2_O_2_ impact during the time-consuming alkylation-fixation procedure and may be not optimal for the alkylation-fixation procedure. When monitored in vivo, fast-reduced H_2_O_2_ indicators can be used to detect H_2_O_2_ variations for small cell populations (for example, for neuronal populations from 10 to 1000 [43,44]); these numbers are limited by the field of view of the fluorescent microscope objective lens. Potentially, if expressed in transgenic mouse lines, slow-reduced NeonOxIrr could allow compilation of a detailed snapshot map for the H_2_O_2_ distribution in neurons of the entire mouse brain. Furthermore, the in vivo observation of H_2_O_2_ using reversible indicators requires attachment of the optical windows that will lead to damage to the skull and brain tissue, and, as a result, to an undesirable inflammatory process in the damaged area, accompanied by the generation of various active oxygen forms, including H_2_O_2_. The combination of the ex vivo alkylation-fixation procedure with NeonOxIrr-expressing transgenic mouse lines may help avoid the undesirable disruption of the tissues of the brain. Despite the optical window limitations, the slow-reduced phenotype of the NeonOxIrr indicator may also be appropriate for the in vivo detection of H_2_O_2_ in animals upon severe impacts, such as irradiation, or the complex behavioral tasks that are incompatible with two-photon imaging with a fixed head.

Our alkylation-fixation protocol with NEM enabled the ex vivo visualization of external H_2_O_2_ in the mouse brain neurons expressing the ratiometric NeonOxIrr-P2A-mCherry indicator (Figure 7b). An advantage of ex vivo visualization of ROS is that samples can be analyzed under identical conditions, which helps avoid artifacts related to variations in pH and other factors. Notably, we failed to detect γ-irradiation-induced ROS production in the brain; this may be due to insufficient sensitivity of the indicator, too high concentration of the indicator as compared with endogenously-produced ROS, non-mitochondrial localization of the indicator, or a slow time course of ROS production upon irradiation. The maximally achievable ex vivo dynamic range for NeonOxIrr in the neurons was limited to 1.5-fold and may be improved in the future by the development of the NeonOxIrr indicator version with the enhanced dynamic range.

A NEM-based alkylation-fixation protocol similar to ours was used for the ex vivo visualization of ROS using Orp1-roGFP and Grx1-roGFP indicators expressed in HeLa, H1975 cells, *Drosophila* larvae, and liver and brain tissues of transgenic mice [7,8,9,45]. However, in cultured neurons and brain tissue, this protocol did not work for the NeonOxIrr indicator, which had higher sensitivity for H_2_O_2_ and a faster oxidation reaction rate than the Orp1-roGFP and Grx1-roGFP indicators. The essential feature of our alkylation-fixation method for neuronal tissue was the use of the NP−40 non-ionic detergent to improve NEM penetration across the neuronal plasma membrane. We assume that our protocol may work for other types of cells and tissues as well, allowing the ex vivo visualization of the H_2_O_2_ distribution in the whole animal brain and other organs in NeonOxIrr transgenes. When applied to diverse biological models, our new indicator may lead to the discovery of new pathways for intracellular H_2_O_2_/ROS production.

## 4. Materials and Methods

### 4.1. Fixation of the NeonOxIrr-P2A-mCherry Indicator in Bacteria Pretreated with Alkylating Reagents

The expression of the NeonOxIrr-P2A-mCherry indicator was performed in BW25113 bacteria in 10 mL of LB medium containing ampicillin (100 μg / mL) and 0.004% arabinose for 24 h at 37 °C, 220 rpm and for 24 h at room temperature. We added 40 mM H_2_O_2_ to a final concentration of 200 μM to 350 μL of bacterial culture and incubated the mixture for 2 min. The bacteria were then pelleted by centrifugation at 13,200 rpm for 1 min, followed by resuspension of the precipitate in 1 mL of an alkylating reagent of NEM or IAM at a concentration of 0–40 mM in PBS buffer, pH 6.5, 7.4, 8.5, or 9.5 followed by incubation at 4, 25, or 37 °C for 10 min. The bacteria were then pelleted by centrifugation at 13,200 rpm for 1 min, followed by resuspension of the precipitate in 1 mL of PBS buffer pH 7.4 and precipitation at 13,200 rpm for 1 min. Fixation was then performed by resuspension of the precipitate in a 1% or 4% solution of paraformaldehyde and subsequent incubation for 20 min at room temperature. The bacteria were then washed twice with PBS by precipitating the bacteria at 13,200 rpm for 1 min and resuspending in 1 mL of PBS. Finally, the pellet of the bacteria was resuspended in 350 μL of PBS. In the case of the alkylation experiment at different temperatures, samples were cooled on ice after hydrogen peroxide treatment at r. t., incubated for 10 min at these temperatures and again stored on ice until a 1% or 4% paraformaldehyde treatment was carried out at room temperature.

### 4.2. Expression in Mammalian Cells 

NeonOxIrr-mCherry fusion proteins were amplified by PCR from bacterial plasmids and inserted into the pLU-CMV-GFP animal vector at the AscI/BsrGI restriction sites (courtesy of Chumakov, Institute of Molecular Biology, Moscow, Russia) (ESI Appendix A). The restriction site of BsrGI was removed from the mCherry protein gene by PCR amplification. pN1-CMV-EGFR-GFP plasmid was from Addgene #32751. The plasmids were transfected into HeLa Kyoto mammalian cells [6] using a TurboFectTM (Thermo Scientific, Vilnius, Lithuania) according to the manufacturer’s protocol. At 24 h after transfection, images of cells were obtained before and after the addition of hydrogen peroxide to a final concentration of 200 μM using a spinning disk Andor WD Technology Revolution multi-point confocal system (Andor Technology Ltd., Belfast, UK) equipped with an inverted Nikon Eclipse Ti microscope, a 75 W mercury-xenon lamp (Hamamatsu Photonics, Hamamatsu, Japan), a 60 × oil immersion objective (Nikon, Tokyo, Japan), a 16-bit Neo sCMOS camera (Andor Technology Ltd., Belfast, UK), laser module Revolution 600 (Andor Technology Ltd., Belfast, UK), spinning-disk module Yokogawa CSU-W1 (Andor Technology, UK), Ti Perfect Focus System enabling automatic focus drift correction over the course of a time-lapse and a cage incubator (Okolab Srl., Pozzuoli, NA, Italy). The green and red fluorescence were acquired using 80% of the 488 nm (17.3 µW/cm2 before objective lens) and 80% of 561 nm (62.3 µW/cm2 before objective lens) laser powers, confocal dichroic mirror 405/488/561/640 and filter wheel emission filters 525/50 and 617/73, respectively. Light power density was measured at a rear focal plane of the objective lens using PM100D power meter (ThorLabs Inc., Newton, NJ, USA) equipped with S120VS sensor (ThorLabs Inc., Newton, NJ, USA).

### 4.3. Alkylation-Fixation of Mammalian Cells and Neuronal Cultures

Alkylation-fixation of a monolayer of HeLa Kyoto cells on 35-mm MatTek glass-bottom dishes was performed 2–3 days after transfection. Alkylation-fixation of neuronal cultures plated on 35-mm MatTek glass-bottom dishes was performed 10–14 days after viral transduction. To this end, cells on 35-mm MatTek glass-bottom dishes were washed 2 times with 2 mL of DPBS buffer (Gibco, Life Technologies Limited, Paisley, UK). The cells were then alkylated by incubation with 1 mL of 1 mM NEM in DPBS buffer for 10 min at room temperature. The alkylated cells were then washed with 2 mL of DPBS buffer and fixed at room temperature for 15–20 min with 2 mL of 4% paraformaldehyde in PBS. The fixed cells were then washed 2 times with 2 mL of DPBS and then stored in DPBS at 4 °C.

### 4.4. Construction of Vectors for the Preparation of Adeno-Associated Virus Particles 

The pAAV-CAG-NeonOxIrr-P2A-mCherry vector was constructed to obtain adeno-associated viral (rAAV) particles for expression in neuronal cultures and in the brains of mice. To this end, the NeonOxIrr-P2A-mCherrry fragment was amplified by PCR as the NheI-BsrGI fragment using the bacterial plasmid pBAD/Myc-HisB-NeonOxIrr-P2A-mCherry as a template and was inserted in place of the fragment iRFP-P2A-mCherry into the vector pAAV-CAG-iRFP-P2A-mCherry.

### 4.5. Preparation of rAAV Particles 

The rAAV particles were purified from ten 15-cm dishes as described previously [46]. 

### 4.6. Isolation, Transduction, and Imaging of Neuronal Cultures

For dissociated neuronal cultures, cells were isolated from C57BL/6 mice at postnatal days 0–3 and were grown on MatTek glass-bottom dishes in Neurobasal Medium A (GIBCO, Life Technologies Limited, Paisley, UK) supplemented with 2% B27 Supplement (GIBCO, UK), 0.5 mM glutamine (GIBCO, UK), 50 U/mL penicillin, and 50 μg/mL streptomycin (GIBCO, UK). On the 4th day in vitro, neuronal cells were transduced with 1 μL rAAV viral particles carrying pAAV-CAG-NeonOxIrr-P2A-mCherry. The cells were imaged using an Andor WD Technology Revolution multi-point confocal system equipped with Ti Perfect Focus System enabling automatic focus drift correction over the course of a time-lapse using the same light powers as described in Section 4.2. 

### 4.7. Introduction of rAAV Particles in the Mouse central Nervous System (CNS) 

Mice were anesthetized by intraperitoneal injection of a mixture of Zoletil (40 mg / kg) and Rometar (5 mg / kg) in 0.9% solution of sodium chloride (0.1 mL per 10 g of weight). Viral particles were introduced with a glass capillary (external diameter at the end of 18–25 µm). For ex vivo studies of H_2_O_2_ generation, CAG-NeonOxIrr-P2A-mCherry rAAV particles were bilaterally injected into the barrel fields of the somatosensory cortex perpendicular to the surface of the skull. The suspension of the viral particles (total volume of 0.2 μL) was injected into one brain hemisphere at the rate of 0.2 μL over 12 min. After the injection of rAAV particles, a 0.7-mm-long cannula for injections (Plastics One) was implanted into the same area.

### 4.8. Oxidation of the Indicator with H_2_O_2_ Followed by its Fixation ex vivo 

Seven days after the surgery, the mice were anesthetized with Zoletil (40 mg/kg) and Rometar (5 mg/kg). Through the cannula, 1 μL of 200 μM H_2_O_2_ solution was introduced into one of the brain hemispheres. Immediately after administration, the mice were given a lethal dose of chloral hydrate (0.25 mL of a 15% solution of chloral hydrate in a 0.9% solution of sodium chloride). In the absence of pain sensitivity, the mice underwent intracardial perfusion, first by 70 mL of 1 mM NEM solution in PBS for 7 min followed by 3 min incubation, and then with 30 mL of 4% paraformaldehyde solution in PBS. After extraction, the animal’s brain passed post-fixation for 18 h in 4% PFA solution at 4 °C. 

### 4.9. Irradiation with γ-radiation

The mice were irradiated at the Kurchatov Institute using a cobalt (Co60) γ-ray source GUT−200M. Experimental non-anesthetized animals were placed near the radiation source in a narrow cage. The absorbed dose of 5 Gy at a power of 1 Gy/min was regulated by the distance from the cage to the radiation source and the exposure time. After irradiation, the mice were returned to their home cage in the vivarium. Thirty-five minutes or 16 h after irradiation, the mice were anesthetized with Zoletil (40 mg/kg) and Rometar (5 mg/kg) and perfused with 70 mL of solution containing 1 mM NEM and 0.01% Nonidet NP40 in PBS for 10 min followed by perfusion with 30 mL of 4% paraformaldehyde (4% PFA) for 3 min at r.t. Brains were then dissected and post-fixed by immersion in 4% PFA overnight at 4 °C. The time from irradiation to alkylation was no more than 35 min for mice with the NeonOxIrr-P2A-mCherry indicator in the cortex and 16 h for mice with the NeonOxIrr-P2A-mCherry indicator in the hippocampus. Sham controls with the NeonOxIrr-P2A-mCherry indicator were treated the same as the irradiated mice but housed in an adjacent room at the background dose rate of 0.013 Rad/hr without being exposed to radiation.

### 4.10. Preparation of Brain Tissue Samples and Visualization

Free-floating slices with a thickness of 50 μm were obtained using vibratome VT1200S (Leica Biosystems, Nussloch, Germany). The slices were mounted on slide glass and cover-slipped over Fluoromount medium (Sigma-Aldrich, Darmstadt, Germany). The resulting slices were imaged using a confocal microscope Andor Revolution WD (Andor Technology Ltd., Belfast, UK) with a 60 × oil immersion objective using the same light powers as described in Section 4.2. The excitation wavelengths of fluorescence were 488 nm for NeonOxIrr and 561 nm for mCherry. The cells located in the area adjoining the end of the cannula were selected for imaging. Later, the bodies of the cells and the area that did not contain cells were isolated in the IQ3 software (Andor Technology Ltd., Belfast, UK); the fluorescence intensities of NeonOxIrr and mCherry along with the value of autofluorescence in the selected field of view were measured in the same software. The intensity ratio for NeonOxIrr to mCherry was calculated according to the following formula:Ratio_488/568_ = (mean_488_ − mean_488 background_)/(mean_568_ − mean_568 background_),
where: mean_488_, mean_488 background_, mean_568_, and mean_568 background_ are the average values of green fluorescence of NeonOxIrr in the body of the cell, the autofluorescence in this field of view in the green channel, the red fluorescence of mCherry in the body of the cell, and the autofluorescence in this field of view in the red channel, respectively. 

### 4.11. Statistical Processing of Data 

Statistical processing of the data was carried out using the Prizm 6 software package (GraphPad). For comparisons between two samples, the Mann-Whitney two-tailed test was used.

## 5. Conclusions

In conclusion, we developed a slowly reducible H_2_O_2_ indicator that can be used for both the in vivo and ex vivo visualization of ROS in cultured HeLa and neuronal cells and in the mouse brain. For in vivo ROS imaging, the NeonOxIrr indicator offers beneficial brightness, pH stability, and sensitivity to both externally applied H_2_O_2_ and optogenetically produced ROS. The NeonOxIrr indicator has a slowly reducible phenotype that enables ex vivo H_2_O_2_ visualization in HeLa cells using the developed alkylation-fixation protocol. A modified version of the alkylation-fixation procedure in the presence of a non-ionic detergent expands the potential use of the NeonOxIrr indicator for ex vivo H_2_O_2_ visualization in neuronal cultures and the mouse brain. Further optimization of the dynamic range of the NeonOxIrr indicator and the replacement of mCherry with other RFPs is likely to yield the ex vivo detection of H_2_O_2_ with substantially improved sensitivity and performance. 

A combination of the transgenic animal lines expressing the NeonOxIrr indicator with the developed alkylation-fixation protocol should facilitate systematic studies of endogenous ROS production during various behavioral training and in response to other conditions. We believe that expanding the family of fixable indicators is a promising strategy to visualize the biological processes at the systems level of the whole organism. 

## Figures and Tables

**Figure 1 ijms-20-03138-f001:**
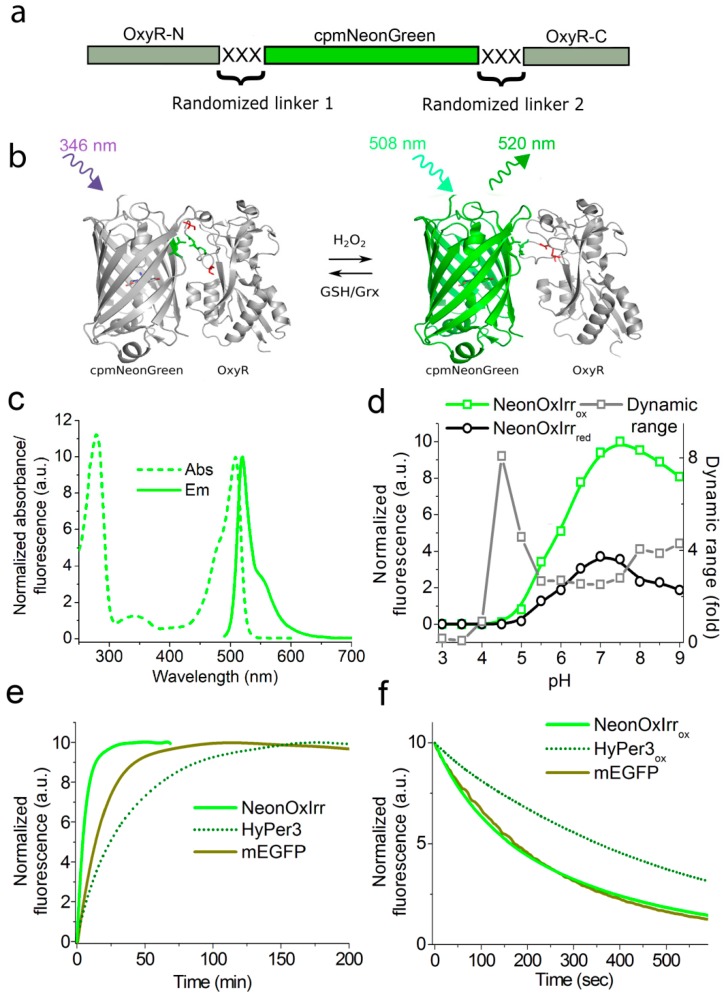
Spectral and biochemical properties of NeonOxIrr. (**a**) Schematic representation of the original library for NeonOxIrr, which consisted of the cpmNeonGreen fluorescent protein inserted into the OxyR sensory part between residues 205 and 206, with 3 randomized amino acid linkers located between the fluorescent and sensory domains. (**b**) Schematic representation of the NeonOxIrr indicator functioning. The cpmNeonGreen fluorescent domain is shown as a β-barrel (PDB 5LTR), which reversibly fluoresces upon oxidation of OxyR cysteines (in red) with H_2_O_2_. The OxyR domain is shown in reduced (PDB 1I69) and oxidized states (PDB 1I6A). The residues attached to the linkers are selected as a green colored ball and stick. Linkers are not shown. Excitation and emission lights are shown with arrows. (**c**) Absorbance and emission spectra of NeonOxIrr in an oxidized state. (**d**) Equilibrium pH dependences for the dynamic range and the fluorescence of NeonOxIrr in the oxidized and reduced (in the presence of 14 mM DTT) states. (**e**) Maturation kinetics of NeonOxIrr, HyPer3 and control mEGFP at 37 °C. (**f**) Photobleaching kinetics for oxidized NeonOxIrr, HyPer3, and mEGFP at a power density of 7.3 mW/cm^2^ at the back focal plane of the objective lens. Proteins were photobleached in aqueous microdroplets in oil. The photobleaching data were normalized to the spectral output of the lamp, transmission profile of the filter and dichroic mirror, and absorbance spectra of the proteins.

**Figure 2 ijms-20-03138-f002:**
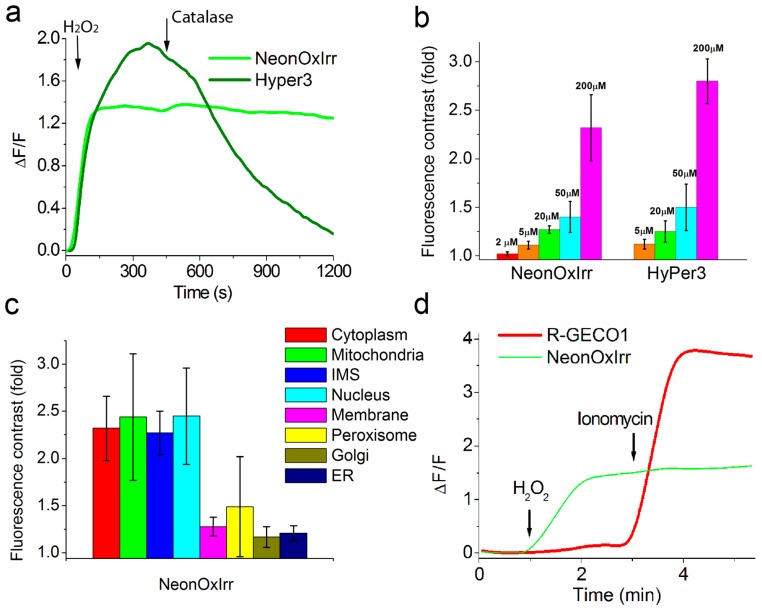
Response of NeonOxIrr to external H_2_O_2_ in the cytoplasm and organelles of live HEK293T cells. (**a**) Dynamics of sensor reactions to the addition of a saturating 200 µM concentration of H_2_O_2_ averaged across 10 cells (two cultures). Times of the addition of H_2_O_2_ (200 µM final) and catalase (5 µg/mL final) are shown with arrows. Dynamics of fluorescence changes were extracted from the series of images acquired on confocal microscope. (**b**) Dependence of maximally achievable contrasts of the NeonOxIrr and HyPer3 indicators from the final concentration of external H_2_O_2_. (**c**) Contrasts of the NeonOxIrr indicator expressed in different compartments of live HEK293T cells. Contrasts correspond to a 200 µM final concentration of H_2_O_2_. (**d**) Simultaneous two-color imaging of H_2_O_2_ and calcium transients in HeLa cells. Averaged response of NeonOxIrr indicator to H_2_O_2_ (200 µM final concentration) related to 2.5 µM ionomycin-induced calcium response of the established R-GECO1 indicator. Error bars are SD.

**Figure 3 ijms-20-03138-f003:**
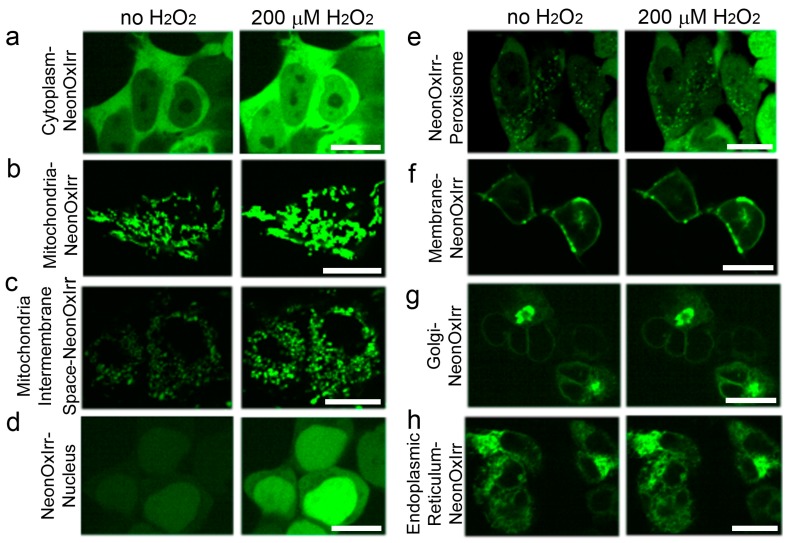
Targeting and response of the NeonOxIrr indicator to external H_2_O_2_ in different compartments of live HEK293T mammalian cells. Confocal images of HEK293T cells transiently expressing the NeonOxIrr indicator in the cytoplasm (**a**), lumen of mitochondria (**b**), intermembrane space of mitochondria (**c**), nucleus (**d**), peroxisomes (**e**), plasma membrane (**f**), Golgi apparatus (**g**), and endoplasmic reticulum (**h**) are shown before and 5 min after H_2_O_2_ addition until a 200 µM final concentration. Scale bars – 20 µm.

**Figure 4 ijms-20-03138-f004:**
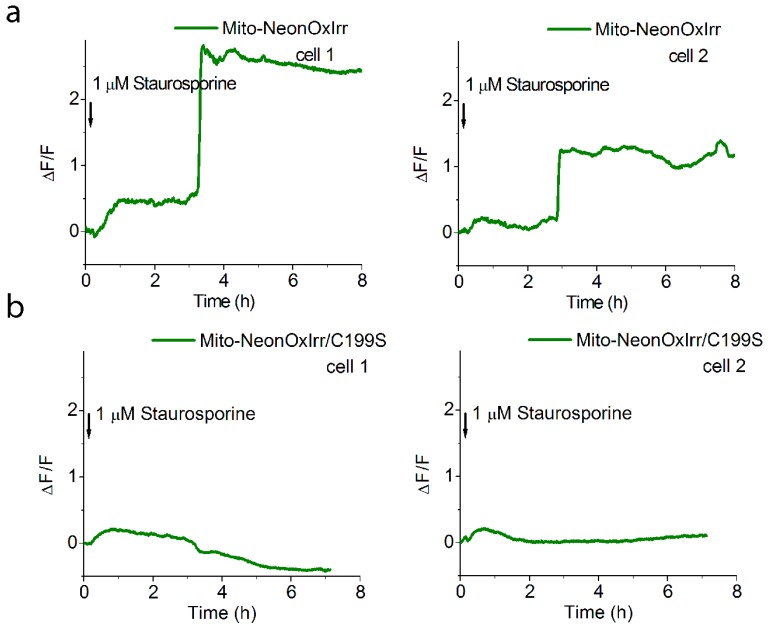
Response of NeonOxIrr to staurosporine-induced endogenous H_2_O_2_ production in the lumen of mitochondria in live HeLa cells. (**a**) Dynamics of green fluorescence of the NeonOxIrr indicator to the addition of staurosporine (1 µM). The responses for three cells from one cell culture are shown. (**b**) Dynamics of green fluorescence of the NeonOxIrr/C199S indicator (with blocked sensitivity to H_2_O_2_) to the consecutive addition of staurosporine (1 µM). The responses for three cells from one cell culture are shown. The time of the staurosporine addition is shown with an arrow. Fluorescence was normalized to the 100% maximal fluorescence achieved for NeonOxIrr after the H_2_O_2_ addition. Dynamics of the fluorescence changes were extracted from a series of images acquired using a confocal microscope.

**Figure 5 ijms-20-03138-f005:**
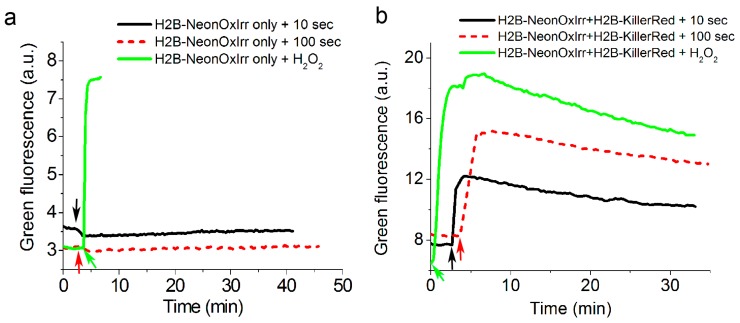
Comparison of the responses of the green NeonOxIrr indicator to external H_2_O_2_ and optogenetic reactive oxygen species (ROS) generated by KillerRed red fluorescent protein that is co-expressed with an indicator in the nucleus of live HEK293T mammalian cells. (**a**) Response of the H2B-NeonOxIrr indicator expressed in the nucleus of HEK293T cells to H_2_O_2_ (green arrow) and yellow light (545/30 nm excitation light power of 3.1 mW/cm^2^ before 60x objective lens) illumination for 10 (black arrow) and 100 s (red arrow). (**b**) Response of H2B-NeonOxIrr co-expressed with H2B-KillerRed in the nucleus of HEK293T cells to H_2_O_2_ (green arrow) and yellow light illumination for 10 (black arrow) and 100 s (red arrow). Black and red plots illustrate changes of green fluorescence for the NeonOxIrr indicator in the nucleus upon illumination with yellow light for 10 and 100 s, respectively. Green graphs show reactions of the NeonOxIrr indicator to external H_2_O_2_ (200 µM final concentration). The time dependences of green fluorescence were extracted from a time-lapse series of fluorescent confocal images of HEK293T cells.

**Figure 6 ijms-20-03138-f006:**
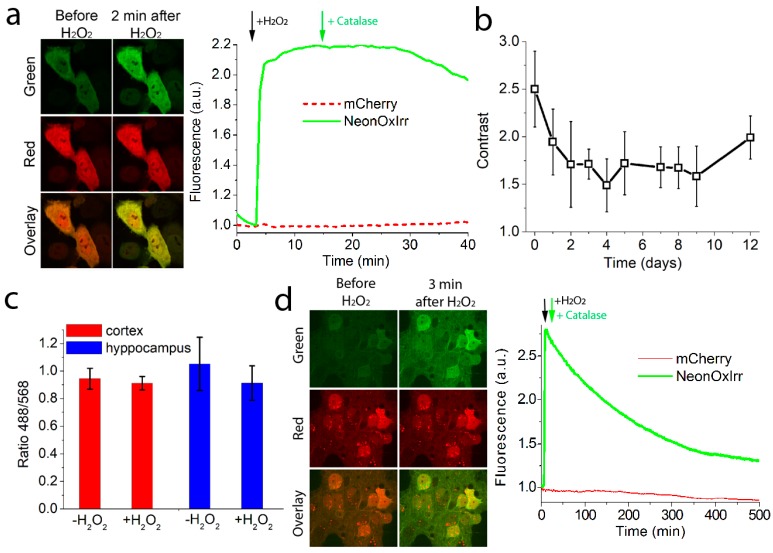
In vivo and ex vivo detection of the external H_2_O_2_ in HeLa mammalian cells and neuronal cultures using green-red NeonOxIrr-P2A-mCherry indicator and N-ethylmaleimide (NEM). (**a**) Confocal images of HeLa cells expressing NeonOxIrr-P2A-mCherry indicator in two channels and their overlay are shown before and 2 min after the addition of 200 μM H_2_O_2_ (panel a, left). Dynamics of the reaction of the NeonOxIrr-P2A-mCherry indicator with H_2_O_2_ in HeLa cells (panel a, right). The black and green arrows correspond to the addition of 200 μM H_2_O_2_ and 5 μg/mL catalase to medium A (20 mM HEPES, pH 7.4, DMEM, 10% FBS, glutamine, penicillin-streptomycin at 37 °C, 5% CO_2_), respectively. (**b**) Time dependence of fluorescence contrast during storage at 4 °C for NeonOxIrr-P2A-mCherry in HeLa cells after alkylation with 1 mM NEM followed by fixation with 4% PFA. (**c**) Green fluorescence normalized to red fluorescence (the 488/568 nm ratio) for neuronal cultures isolated from the cortex or hippocampus of the mice brains (C57 line) in which H_2_O_2_ was injected (+ H_2_O_2_) or not (-H_2_O_2_) for 5 min followed by alkylation with 1 mM NEM for 10 min and 4% PFA fixation for 20 min at r.t. (**d**) Confocal images of neuronal cultures (DIV 14) expressing NeonOxIrr-P2A-mCherry indicator in two channels and their overlay are shown before and 3 min after the addition of 200 μM H_2_O_2_. Neuronal cultures at DIV 4 were transduced with rAAV particles carrying CAG-NeonOxIrr-P2A-mCherry. Dynamics of the reaction of the NeonOxIrr-P2A-mCherry indicator with external H_2_O_2_ in neuronal culture (panel d, right). The black and green arrows correspond to the addition of 200 μM H_2_O_2_ (5 min) and 5 μg/mL catalase (9.7 min) to medium A, respectively.

**Figure 7 ijms-20-03138-f007:**
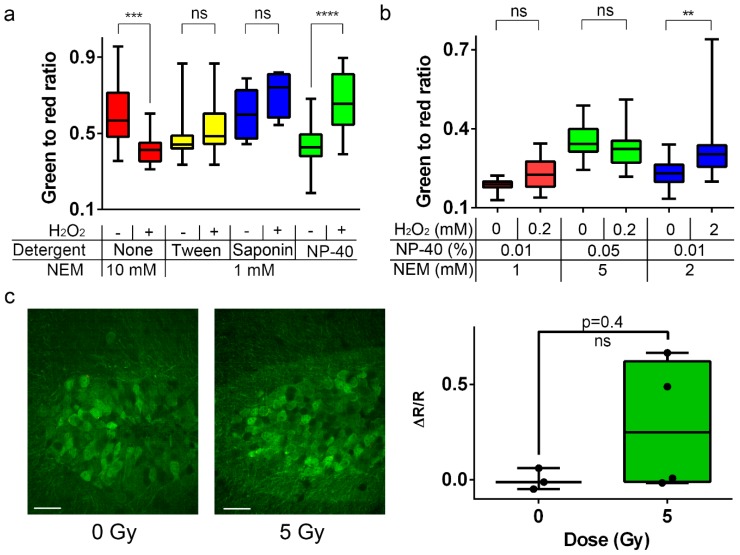
Ex vivo detection of external H_2_O_2_ or γ-irradiation-induced ROS on neuronal cultures and mouse brain cortex and hippocampus using NeonOxIrr-P2A-mCherry indicator and NEM with detergents. (**a**) Ex vivo dependence of the H_2_O_2_-induced contrast of the NeonOxIrr-P2A-mCherry indicator in neuronal cultures. Various detergents have been added during the alkylation with NEM, which was followed by fixation with PFA. The green fluorescence of the NeonOxIrr-P2A-mCherry indicator normalized to the red fluorescence of mCherry (ratio 488/561) in neuronal cultures (DIV 16) after alkylation with 10 or 1 mM NEM in the presence of 0.1% Saponin, 0.01% NP−40, or 0.1% Tween−20 and subsequent fixation with 4% PFA. The cultures were pre-incubated with external 200 μM H_2_O_2_ or without H_2_O_2_ for 5 min. The corresponding contrasts are shown on the 4th day after the alkylation-fixation procedure. (**b**) Ex vivo detection of the NeonOxIrr-P2A-mCherry indicator response to external H_2_O_2_ in the mouse cortex. The averaged values of green fluorescence normalized to the red fluorescence (ratio 488/561) are shown for neuronal cells from confocal images of mouse brain slices without the injection of H_2_O_2_ and with the injection of 0.2 or 2 mM H_2_O_2_. Brain slices were obtained after the extraction of brains from perfused animals. Perfusion of slices pre-alkylated with 1, 2, or 5 mM NEM for 10 or 40 min (the last two green bars) was performed with 4% PFA. Immediately prior to alkylation-perfusion, 0.2 mM or 2 mM H_2_O_2_ was injected into only one of the two regions infected with rAAV particles bearing NeonOxIrr-P2A-mCherry. Images of brain slices were obtained on the first or second day after fixation. (**c**) Ex vivo detection of the NeonOxIrr-P2A-mCherry indicator response 16 h after 5 Gray γ-irradiation in the hippocampus of the mice brain. Confocal images of brain hippocampal slices transduced with rAAV-CAG-NeonOxIrr-P2A-mCherry viral particles are shown for the control mice (0 Gy dose) and 5 Gy γ-irradiated mice (panel c, left). Scale bar—30 µm. ΔR/R values are shown for the neuronal cells from confocal images of slices of mouse hippocampus regions without and with 5 Gy γ-irradiation. Mean values of the green fluorescence were normalized to the red fluorescence (R, ratio 488/561). Brain slices were obtained after the extraction of brains from perfused animals. Perfusion of slices pre-alkylated with 1 mM NEM for 10 min was performed with 4% PFA 16 h after irradiation. The box extends from the 25th to 75th percentiles. The line in the middle of the box is plotted at the median. The whiskers go down to the smallest value and up to the largest.

**Table 1 ijms-20-03138-t001:** In vitro characteristics of the NeonOxIrr indicator compared to HyPer3 and mEGFP.

Protein	NeonOxIrr		HyPer3		mEGFP
	Red	Ox	Red	Ox	
Absorbance maximum (nm)	508		498		490
Extinction coefficient (M^−1^cm^−1^)^1^	ND	110,000	ND	65,000	64,000
Emission maximum (nm)	520		518		512
Quantum yield^2^	0.57		0.17		0.60
Relative brightness (%)	ND	170	ND	29	100
p*K_a_*^3^	5.9	5.9	≥7.9	≥7.6	5.9
pH-range (pH changes impact to contrast)^4^	pH 5.5–8.5 (6–52%)		pH 5.5–8.5 (62–596%)		NA
Maturation t_0.5_ at 37 °C (min)^5^	4.2		25		14
Photobleaching t_0.5_ (s)^6^	168 ± 30		354 ± 3		174 ± 8
Fluorescence contrast on bacterial suspension (fold)^7^	2.8 ± 0.6		2.5 ± 0.5		NA

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
