# Peer review of "Slowly Reducible Genetically Encoded Green Fluorescent Indicator for In Vivo and Ex Vivo Visualization of Hydrogen Peroxide"

_ijms, 2019, doi:10.3390/ijms20133138_

Round 1
Reviewer 1 Report
In this manuscript, Subach and colleagues report on the development of a novel fluorescent indicator for the detection of hydrogen peroxide. Particularly in the field of oxidative stress research, but also for monitoring cellular metabolism and vitality, an imaging probe for detecting H2O2 is highly desirable. A fluorescent imaging probe based on green fluorophores would allow for a rather borad applications. The indicator developed by the authors, NeonOxIrr, offers several advantageous properties compared to current state-of-the-art indicators. For the first time, an in vivo application seems within reach. The biophysical characterization of the new indicator is straightforward and rather comprehensive. What is more, the authors apply the indicator both in non-neuronal cells (HeLa) as well as in primary neuronal cultures and post mortem tissue preparations. Generally, the manuscript is well-written. While I support the notion of the authors, that this indicator might represent an important step forward in terms of H2O2 indicators, the manuscript does not convey a clear picture on the capabilities and shortcomings of the indicator. These are my concerns:
1. The authors claim, already in the title, that the new indicator might be suitable for in vivo applications. In the manuscript, the authors guide the reader through various protocols, how eventually at least an application in neuronal cell cultures seems doable. The results section is quite convoluted, and should be re-organized, so that the reader can get a clear picture on the final protocol. Ideally, the authors should provide a scheme on the best application protocols for the possible applications (HeLa, neuronal cultures, cortical slices, and in vivo).
2. The confocal images shown in Figure 3 are not acceptable: The authors need to provide an independent marker for fluorescence emission, such as a quantum dot, so that differences in emission can be better appreciated. This should be related to established in vivo indicators, e.g. of calcium, to judge whether in vivo applications might be achievable.
3. Along those lines, the authors need to expand the description of the imaging hardware needed for visualization in vitro, ex vivo, and potentially in vivo. Which illumination power, and which detector, emission sensitivity is needed?
4. Figure 4, why did the authors use “normalized fluorescence” in arbitrary units? They should employ the standard in the field, and show df/f0, so that the reader can compare the relative increase to other indicators in the field.
5. Line 565, the authors state, that the NeonOxIrr-P2A-mCherry version does not work for neuronal cultures. I certainly agree, that it is important to share negative results, but, as mentioned above, the results section seems rather to reflect a compendium of tried methods. This needs to be re-structured.
6. Figure 7 c, there are no scale bars, no normalization of the intensities, and again, arbitrary units in the quantification.
7. All bar graphs should be converted to box-whisker plots, to better grasp the data distribution.
Author Response
Response to Reviewer 1 Comments
We thank reviewer 1 for his valuable comments and useful suggestions, which we have addressed entirely in the revised manuscript. As requested by the reviewer 1 we added additional experiments that were performed to further characterize the novel NeonOxIrr hydrogen peroxide indicator. In particular, we carried out additional experiments to compare the responses of the NeonOxIrr redox indicator and established R-GECO1 calcium indicator in cultured cells.
Overall, the revised manuscript has 6 new display items such as the Figures 2d, S6, S7, and S9, Scheme S1 and Video S1. In the main text we also moved two chapters with negative results to Supplementary Results.
Reviewer #1:
In this manuscript, Subach and colleagues report on the development of a novel fluorescent indicator for the detection of hydrogen peroxide. Particularly in the field of oxidative stress research, but also for monitoring cellular metabolism and vitality, an imaging probe for detecting H2O2 is highly desirable. A fluorescent imaging probe based on green fluorophores would allow for a rather borad applications. The indicator developed by the authors, NeonOxIrr, offers several advantageous properties compared to current state-of-the-art indicators. For the first time, an in vivo application seems within reach. The biophysical characterization of the new indicator is straightforward and rather comprehensive. What is more, the authors apply the indicator both in non-neuronal cells (HeLa) as well as in primary neuronal cultures and post mortem tissue preparations. Generally, the manuscript is well-written. While I support the notion of the authors, that this indicator might represent an important step forward in terms of H2O2 indicators, the manuscript does not convey a clear picture on the capabilities and shortcomings of the indicator. These are my concerns:
Point 1: The authors claim, already in the title, that the new indicator might be suitable for in vivo applications. In the manuscript, the authors guide the reader through various protocols, how eventually at least an application in neuronal cell cultures seems doable. The results section is quite convoluted, and should be re-organized, so that the reader can get a clear picture on the final protocol. Ideally, the authors should provide a scheme on the best application protocols for the possible applications (HeLa, neuronal cultures, cortical slices, and in vivo).
Response 1: In the revised manuscript, we moved two chapters with negative results (2.10. Ex vivo H2O2 detection failure in neuronal cultures expressing ratiometric NeonOxIrr-P2A-mCherry and 2.11. Ex vivo H2O2 detection failure in the mouse brain expressing ratiometric NeonOxIrr-P2A-mCherry) to Supplementary Material section, Supplementary Results.
In the revised manuscript, Supplementary Material section, we added Scheme S1, describing the recommended alkylation-fixation protocols for the possible applications.
Point 2: The confocal images shown in Figure 3 are not acceptable: The authors need to provide an independent marker for fluorescence emission, such as a quantum dot, so that differences in emission can be better appreciated. This should be related to established in vivo indicators, e.g. of calcium, to judge whether in vivo applications might be achievable.
Response 2: We agree with the reviewer that difference in emission can be better appreciated using quantum dots. In accordance to this issue we alternatively suggested ratiometric green/red version of NeonOxIrr indicator, NeonOxIrr-P2A-mCherry. For the more accurate estimation of fluorescence changes of NeonOxIrr indicator, the changes in its fluorescence in green channel were estimated vs fluorescence of permanently fluorescent mCherry red fluorescent protein in red channel (Figure 6a,d). Note that during short-term confocal imaging with automatic adjustment of the focus we practically did not see the fluorescence changes in control red channel and shift of the cells from the focus (Figure 6a,d).
In this respect in the revised manuscript, Materials and Methods section, we added “... confocal system equipped with Ti Perfect Focus System enabling automatic focus drift correction over the course of a time-lapse.”.
We want to keep Figure 3 because it shows visual response and localization of the NeonOxIrr indicator targeted to the different compartments of the cells. Also we note that it is common practice in the field of genetically encoded indicators not to use references for monitoring of short-term fluorescence changes, e.g. in such high impact journal as Science (Zhao et al. Science, 2011).
To relate NeonOxIrr indicator with established in vivo indicator such as R-GECO1 calcium indicator, in the revised manuscript, we added new Figure 2d, ESI Figure S7 and ESI Video S1. In the main text we added “We next related the response of NeonOxIrr indicator to commonly used R-GECO1 calcium indicator and demonstrated simultaneous two-color imaging of H2O2 and calcium transients in mammalian cells. We transiently co-expressed NeonOxIrr hydrogen peroxide indicator and R-GECO1 calcium indicator in HeLa cells. The response of NeonOxIrr to 200 µM H2O2 was significantly 3-fold lower as compared with ionomycin-induced calcium response for the commonly used R-GECO1 calcium indicator (Figure 2d, ESI Figure S7 and ESI Video S1). Therefore, in combination with R-GECO1 calcium indicator, NeonOxIrr indicator allows simultaneous monitoring of H2O2 and calcium transients in mammalian cells using two-color confocal microscopy.” and “Hence, when expressed in the cytoplasm of mammalian cells, NeonOxIrr demonstrated a 10-fold higher brightness, 1.4-fold lower dF/F response and 7.6-fold longer reduction time compared to control HyPer3 hydrogen peroxide indicator and 3-fold lower response compared to established R-GECO1 calcium indicator.”
Point 3: Along those lines, the authors need to expand the description of the imaging hardware needed for visualization in vitro, ex vivo, and potentially in vivo. Which illumination power, and which detector, emission sensitivity is needed?
Response 3: In the revised manuscript, Materials and Methods section 4.2 we added “The green and red fluorescence were acquired using 80% of the 488 nm (17.3 µW/cm2 before objective lens) and 80% of 561 nm (62.3 µW/cm2 before objective lens) laser powers, confocal dichroic mirror 405/488/561/640 and filter wheel emission filters 525/50 and 617/73, respectively. Light power density was measured at a rear focal plane of the objective lens using PM100D power meter (ThorLabs, Germany) equipped with S120VS sensor (ThorLabs, Germany). ”
In the revised manuscript, Materials and Methods section 4.2 we mentioned “...equipped with ... a 16-bit Neo sCMOS camera (Andor Technology, UK), laser module Revolution 600 (Andor Technology, UK), spinning-disk module Yokogawa CSU-W1 (Andor Technology, UK),... ”
In the revised manuscript, ESI, Supplementary Materials and Methods, Mutagenesis and screening of libraries section, we added “Green fluorescence was registered by 480/40BP excitation (75 µW/cm2 on the sample)...”
For photobleaching experiments we provided light power density before objective lens (7.3 mW/cm2) (please, see ESI, Supplementary Materials and Methods section, Protein purification and characterization section.).
In the revised manuscript, legend to Figure 5, we added “...and yellow light (545/30 nm excitation light power of 3.1 mW/cm2 before 60x objective lens) illumination ...”
Point 4: Figure 4, why did the authors use “normalized fluorescence” in arbitrary units? They should employ the standard in the field, and show df/f0, so that the reader can compare the relative increase to other indicators in the field.
Response 4: In the revised manuscript, Figure 4, we corrected “normalized fluorescence” and show dF/F.
Point 5: Line 565, the authors state, that the NeonOxIrr-P2A-mCherry version does not work for neuronal cultures. I certainly agree, that it is important to share negative results, but, as mentioned above, the results section seems rather to reflect a compendium of tried methods. This needs to be re-structured.
Response 5: In the revised manuscript, we moved two chapters with negative results (2.10. Ex vivo H2O2 detection failure in neuronal cultures expressing ratiometric NeonOxIrr-P2A-mCherry and 2.11. Ex vivo H2O2 detection failure in the mouse brain expressing ratiometric NeonOxIrr-P2A-mCherry) to Supplementary Material section, Supplementary Results.
Point 6: Figure 7 c, there are no scale bars, no normalization of the intensities, and again, arbitrary units in the quantification.
Response 6: In the revised manuscript, Figure 7c, we added scale bars and normalized the green-to-red intensities ratios.
Point 7: All bar graphs should be converted to box-whisker plots, to better grasp the data distribution.
Response 7: In the revised manuscript, Figure 7a-c, we converted bar graphs to box-whisker plots.

Reviewer 2 Report
The manuscript by Subach et al. presents the development and characterization of new fluorescent indicator for hydrogen peroxyde. The article describes the design and screening of mutant libraries based on the cpmNeongreen FP that yielded a clone that is next further characterized. The clone, named NeonOxIrr, was first compared in vitro for its photophysical properties in comparison to HyPer3, a standard H2O2 indicator, and mEGFP. This analysis highlighted the significantly slower reduction rate of the indicator. The indicator was next evaluated in mammalian cells for external H2O2, endogenously and optogenetically produced ROS. The authors then engineered and characterized a ratiometric system with mCherry as a ROS-insensitive red reporter and optimized fixation conditions to benefit from the slow reduction of the probe. The system was finally evaluated for ex-vivo imaging of H2O2 in cell lines and neurons (culture and animal) in response to various stimuli. Overall, the new probe represents a significant progress over existing ones, in particular for its brightness, pH stability and slow kinetics in particular for the applications that they allow such as ex-vivo detection. However, as openly presented and discussed by the authors, the probe still present some limits that could be the object of future improvements.
The article is clearly written and the experiments very well presented. The main text is complemented with an extensive and very detailed supplementary file. The data supports the conclusions and importantly the limits of the new tool have been critically presented and discussed. Even though due to its limitation, the new probe is not as impressive as the initial in vitro properties could have suggested, it still opens up new applications in the field. Taken together with the overall quality of the manuscript both in content and format, I think that this work is suitable for publication.
MInor comments:
-Very few typos detected.
-It is not clear to me why the authors claim that their probe oxidized faster than HyPer3 (l278). When comparing Fig2a and table S3, it seems that the results are contradictory.
-Why would fusion to vimentin decrease the cytoplasmic fraction (L370)
Author Response
Response to Reviewer 2 Comments
We thank reviewer 2 for his valuable comments and useful suggestions, which we have addressed entirely in the revised manuscript. As requested by the reviewer 2 we added additional experiments that were performed to further characterize the novel NeonOxIrr hydrogen peroxide indicator. In particular, we assessed expression levels of NeonOxIrr alone and in fusion with vimentin protein in mammalian cells.
Overall, the revised manuscript has 6 new display items such as the Figures 2d, S6, S7, and S9, Scheme S1 and Video S1. In the main text we also moved two chapters with negative results to Supplementary Results.
Reviewer #2:
The manuscript by Subach et al. presents the development and characterization of new fluorescent indicator for hydrogen peroxyde. The article describes the design and screening of mutant libraries based on the cpmNeongreen FP that yielded a clone that is next further characterized. The clone, named NeonOxIrr, was first compared in vitro for its photophysical properties in comparison to HyPer3, a standard H2O2 indicator, and mEGFP. This analysis highlighted the significantly slower reduction rate of the indicator. The indicator was next evaluated in mammalian cells for external H2O2, endogenously and optogenetically produced ROS. The authors then engineered and characterized a ratiometric system with mCherry as a ROS-insensitive red reporter and optimized fixation conditions to benefit from the slow reduction of the probe. The system was finally evaluated for ex-vivo imaging of H2O2 in cell lines and neurons (culture and animal) in response to various stimuli. Overall, the new probe represents a significant progress over existing ones, in particular for its brightness, pH stability and slow kinetics in particular for the applications that they allow such as ex-vivo detection. However, as openly presented and discussed by the authors, the probe still present some limits that could be the object of future improvements.
The article is clearly written and the experiments very well presented. The main text is complemented with an extensive and very detailed supplementary file. The data supports the conclusions and importantly the limits of the new tool have been critically presented and discussed. Even though due to its limitation, the new probe is not as impressive as the initial in vitro properties could have suggested, it still opens up new applications in the field. Taken together with the overall quality of the manuscript both in content and format, I think that this work is suitable for publication.
MInor comments:
Point 1: -Very few typos detected.
Response 1: We thank the reviewer for the careful reading of our manuscript.
Point 2: -It is not clear to me why the authors claim that their probe oxidized faster than HyPer3 (l278). When comparing Fig2a and table S3, it seems that the results are contradictory.
Response 2: In the revised manuscript, Figure 2a, we replaced traces for individual cells with new traces averaged across 10 cells.
In the revised manuscript, in Supplementary Materials section, we added Figure S6 showing the dF/F response and oxidation rate constants for NeonOxIrr and HyPer3 indicators at different H2O2 concentrations.
Point 3: -Why would fusion to vimentin decrease the cytoplasmic fraction (L370)
Response 3: To address this issue, we compared the brightness of NeonOxIrr and Vimentin-NeonOxIrr fusion relative to co-expressed mCherry protein in HeLa cells. Under the same imaging conditions the normalized brightness of vimentin-targeted NeonOxIrr was the same as compared to the brightness of cytoplasmically-localized NeonOxIrr (ESI Figure S9). Taken into account that fluorescence intensity of NeonOxIrr is proportional to the amount of the protein we assumed that fusion to vimentin did not affect its cytoplasmic fraction.
In the revised manuscript, we added ESI Figure S9 and moved description of vimentin-NeonOxIrr fusion to the chapters 2.3 and 2.4.